# Descent trajectory reconstruction and landing site positioning of Chang'E-4 on the lunar farside

Jianjun Liu[1,2], Xin Ren [1], Wei Yan [1], Chunlai Li [1,2], He Zhang [3], Yang Jia[3], Xingguo Zeng[1], Wangli Chen[1], Xingye Gao[1], Dawei Liu[1], Xu Tan[1], Xiaoxia Zhang[1], Tao Ni[1,2], Hongbo Zhang[1], Wei Zuo [1], Yan Su[1] & Weibin Wen[1]

Chang'E-4 (CE-4) was the first mission to accomplish the goal of a successful soft landing on the lunar farside. The landing trajectory and the location of the landing site can be effectively reconstructed and determined using series of images obtained during descent when there were no Earth-based radio tracking and the telemetry data. Here we reconstructed the powered descent trajectory of CE-4 using photogrammetrically processed images of the CE-4 landing camera, navigation camera, and terrain data of Chang'E-2. We confirmed that the precise location of the landing site is 177.5991°E, 45.4446°S with an elevation of −5935 m. The landing location was accurately identified with lunar imagery and terrain data with spatial resolutions of 7 m/p, 5 m/p, 1 m/p, 10 cm/p and 5 cm/p. These results will provide geodetic data for the study of lunar control points, high-precision lunar mapping, and subsequent lunar exploration, such as by the Yutu-2 rover.

[1] Key Laboratory of Lunar and Deep Space Exploration, National Astronomical Observatories, Chinese Academy of Sciences, Beijing 100101, China. [2] School of Astronomy and Space Science, University of Chinese Academy of Sciences, Beijing 100049, China. [3] China Academy of Space Technology, Beijing 100094, China. Correspondence and requests for materials should be addressed to C.L. (email: licl@nao.cas.cn)

The Chang'E-4 (CE-4) spacecraft successfully landed on the lunar farside on January 3, 2019. The planned durations of the scientific investigations for the lander and the Yutu-2 rover are 6 and 3 months, respectively. After launch on December 8, 2018, CE-4 conducted several stages, including transfer from the Earth to the Moon, orbiting around the Moon and a powered descent. The powered descent stage was the most important stage for soft landing and was implemented under complete autonomous control[1]. According to the orbit design, a 7500 Newton variable propulsion engine was used for deceleration during the CE-4 powered descent stage, which involved phases of main braking, attitude adjusting, vertical descent, hover and hazard-avoidance, slow descent[1–5]. To better understand the autonomous control result and to service orbit control strategy analysis and detection mission planning for subsequent missions, it is of great significance to accurately reconstruct the landing trajectory and determine the location of the landing site after a safe landing.

Radio measurements[6–12], real-time telemetry data[2,13], and image-based matching[3,14–19] are commonly used for powered descent trajectory reconstruction and landing site positioning on a planetary surface. Because there was no radio measurement equipment between the CE-4 and the relay satellite Queqiao, it was unable to perform direct or indirect radio measurements by the ground tracking network on the lunar farside. In addition, the telemetry data received by Queqiao (including the detector altitude, acceleration, and attitude) were not released and could not to be used. As a result, it is difficult to accurately reconstruct the spacecraft's landing trajectory and to confirm the precise landing site location. However, these problems can be effectively solved through localization technology based on landing images, which is not affected by factors such as the lunar gravity field and the dynamical model.

In this study, high-frequency landing sequence images transmitted by Queqiao after safe landing were used for descent trajectory reconstruction and landing site positioning of CE-4, which completely recorded the entire process of the powered descent. We reconstructed the descent trajectory, showing even barely perceivable maneuvers of the spacecraft during the landing approach and that the landing site can also be precisely localized. Here, the digital orthophoto map (DOM) and digital elevation model (DEM) of Chang'E-2 (CE2TMAP2015, see the data descriptions of the Methods section[20,21]) were used as the geographical reference data. The CE-4 landing camera (LCAM, see the Methods section for details) sequence images were applied to reconstruct the CE-4 powered descent trajectory by the photogrammetry bundle adjustment technology[22,23] (see the Methods for a detailed technical flow). Using combined binocular stereoscopic images obtained by the navigation camera (NCAM, see the Methods), which consists of two cameras, we confirmed that the precise location of the landing site is 177.5991°E, 45.4446°S with an elevation of −5935 m.

## Results

### Descent trajectory from LCAM images

The sequence images obtained by the LCAM in vertical attitude after the attitude adjusting phase of CE-4 were used for the reconstruction of the landing trajectory in this study.

According to the reconstructed trajectory, CE-4 adopted a vertical descent approach to gradually approach the landing zone after the attitude adjusting phase with an altitude of 5635 m and a velocity of ~85 m/s. During the period of descending to an altitude of 4130 m above the lunar surface, the trajectory moved 77 m to the north and crossed a crater with a diameter of approximately 200 m. CE-4 reached over area A, as indicated in Fig. 1, with a velocity of ~78 m/s. Over area A, CE-4 decreased

from 4130 m from the lunar surface to 1495 m, the velocity decreased to ~51 m/s, and the trajectory was adjusted from the northeast; there are some craters with diameters of ~70–100 m to the northwest, where is more flat (Fig. 1d, e). Then, the trajectory moved 244 m to the northwest from the altitude of 1495 m to 99 m and the velocity decreased to 0 m/s. CE-4 reached over area B in Fig. 1 and entered the hover and hazard-avoidance phase.

At an altitude of 99 m above the lunar surface (B in Fig. 1), CE-4 hovered for ~13 s. Then, the trajectory moved 12 m to the southwest and crossed the crater with a diameter of 25 m. The altitude above the lunar surface decreased to 30 m, and the velocity changed to ~1.5 m/s. Finally, the CE-4 trajectory slowly descended vertically and, the spacecraft landed safely.

### Landing site localization based on multisource image data

We generated LCAM topographic maps (including the DOM and DEM data) with different spatial resolutions by photogrammetry bundle adjustment technology[22,23]. These maps used a total of 180 images from the LCAM sequence images below an altitude of 5635 m (one image per second), twelve horizontal and vertical control points (GCPs), and one vertical GCP selected on the CE2TMap2015 (Fig. 2a). Using the stereoscopic images obtained by the NCAM from the Yutu-2 rover on the top of the lander and a total of 14 GCPs selected from the LCAM DOM and DEM (Fig. 2b), the DOM with a resolution of 2 cm/p was produced by the close-range photogrammetry technology. In this way, the NCAM DOM was precisely registered to the LCAM DOM. According to the position relationship between the NCAM and the origin of the lander body coordinate system, the horizontal position and elevation of the lander in the LCAM terrain data were measured, which were determined as the final position of the CE-4 landing site (see the Methods). Detailed information on the location of the landing site is shown in Table 1 and Fig. 3.

We also marked the location of the landing site on the LCAM images and CE-2 DOM subdivision with a resolution of 7 m/p (Table 1 and Fig. 4).

The images of the LCAM and Yutu-2 panoramic camera (PCAM) (Fig. 5) show that CE-4 landed on a gentle slope of a degraded crater, only 8.35 m from the rim of a 25 m crater to the north and was surrounded by 5 craters with diameters of 10.21~25.00 m and depths of 1.21~3.03 m. CE-4 landing site can be also located in the Lunar Reconnaissance Orbiter (LRO) narrow angle camera (NAC) images, whose relative position to the surrounding terrain is consistent with our results in Fig. 5.

## Discussion

Similar to Chang'E-3 (CE-3), three types of sensors were employed by CE-4, including a laser ranging sensor, a microwave ranging and velocimeter sensor and an optical imaging sensor and a laser imaging sensor[1]. The powered descent process was realized by autonomous control technology with adaptive powered explicit guidance, multibeam fault-tolerant navigation, and partition quaternion control. However, the trace control strategies for CE-3 and CE-4 are different, and this difference is also reflected in the trajectory recovered in this article.

The CE-3 landing site is located on the northern Mare Imbrium of the lunar nearside. The length of the CE-3 nadir flight trace is ~450 km from south to north. The terrain along the nadir flight trace shows a flat, small variation, gradually decreasing from south to north with a maximum variation of ~800 m (Fig. 6). In the main braking phase, ranging measurements were introduced to correct the bias and drift of the inertial measurement unit (IMU) in time and ensure the stability of the powered descent trajectory. At an altitude of 3~2.4 km, the CE-3 lander performed quick attitude adjustments to a vertical attitude.

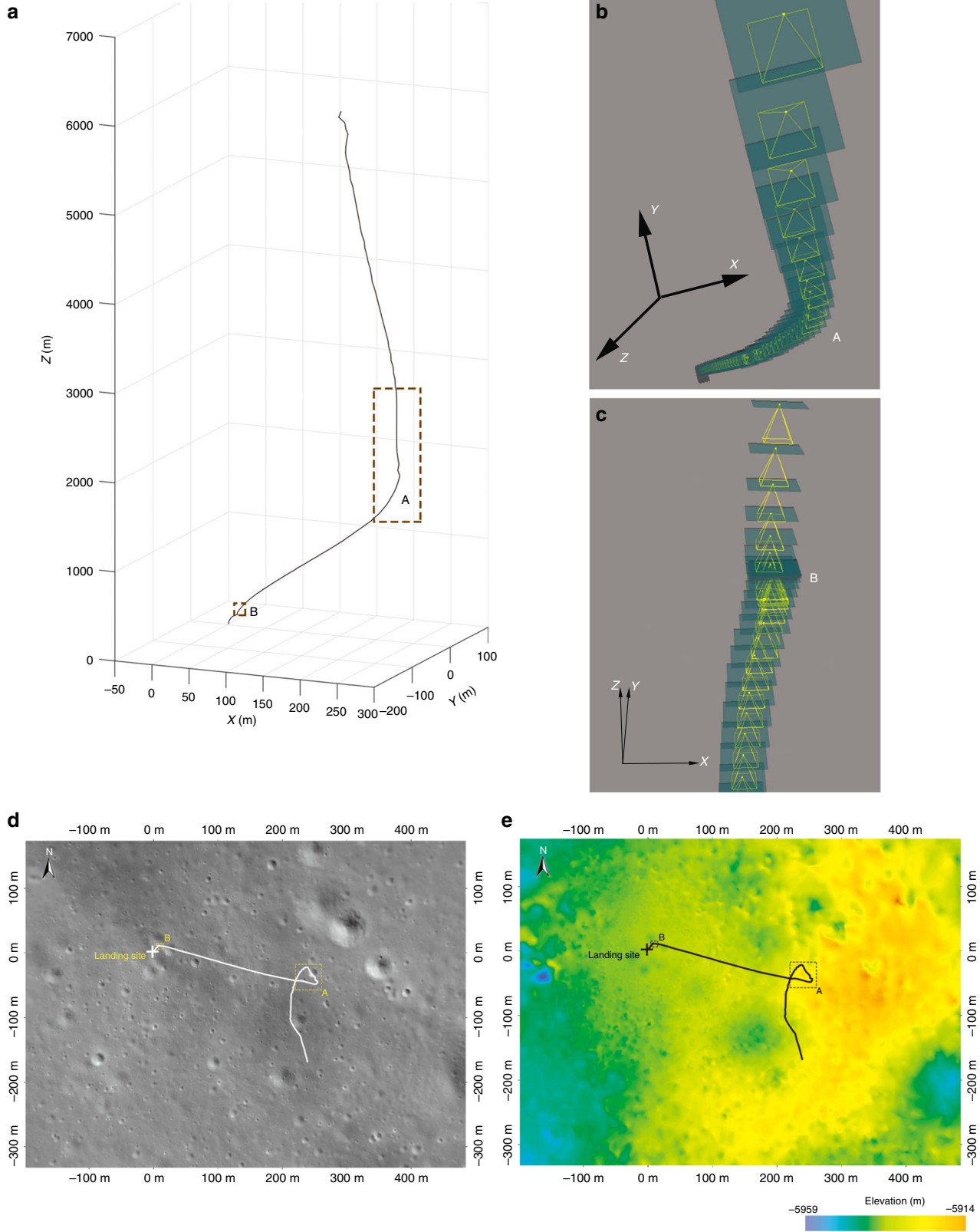

**Fig. 1** CE-4 powered descent trajectory. The + is the identified landing location. **a** CE-4 descent trajectory from the altitude of 6000 m to the lunar surface; **b**, **c** are zoomed images of the descent trajectories of A and B, respectively. The green box represents the position of the LCAM focal plane, and the yellow cone represents the field of view of the LCAM. **d** The solid white line is the projection of the descent trajectory on the LCAM DOM. **e** The solid black line is the projection of the descent trajectory on the LCAM DEM. The coordinate system in this figure is the tangent plane coordinate system of the landing site. This system uses the location of the landing site as the coordinate origin, with its X and Y axes pointing towards the geographical east and the north of the landing site, respectively. The Z-axis, X-axis, and Y-axis form a right-hand system, with the Z-axis pointing to the zenith direction of the landing site

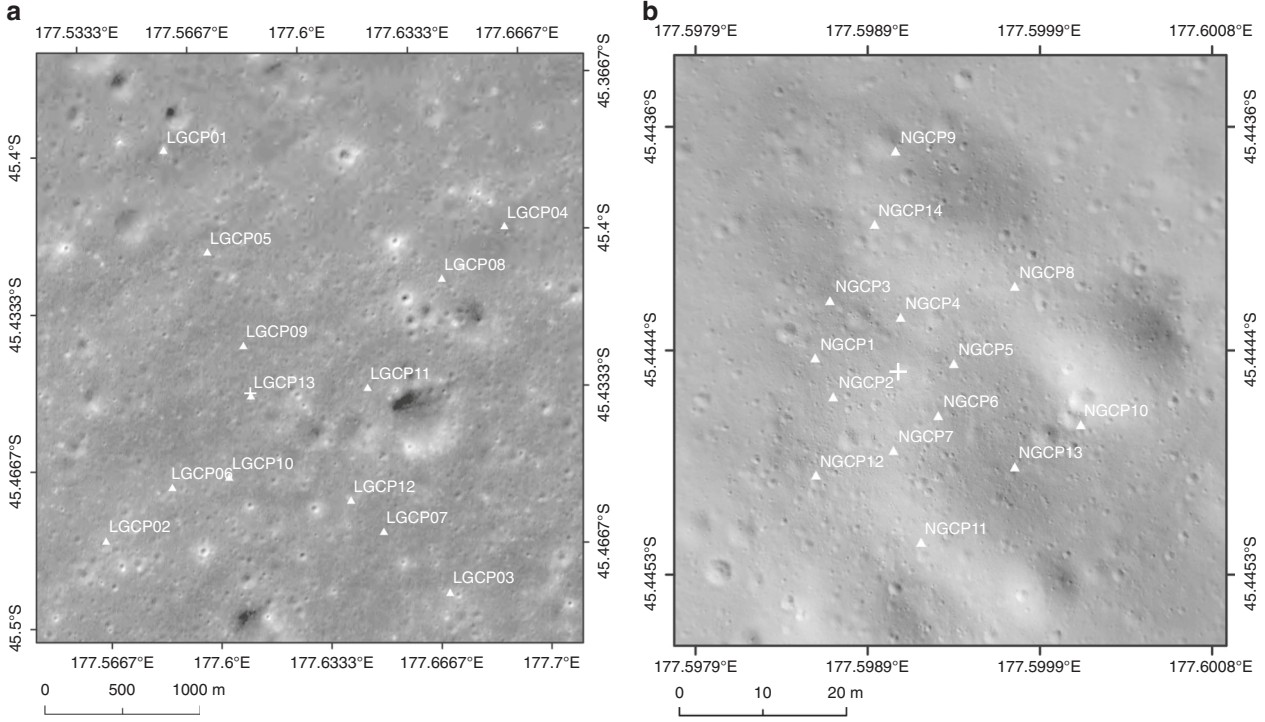

**Fig. 2** GCPs map for photogrammetric adjustment processing. **a** GCPs on the CE2TMap2015 DOM for the LCAM image adjustment, marked as LGCPs, in which LGCP13 is the vertical GCP; **b** GCPs on the LCAM DOM for the NCAM image adjustment, marked as NGCPs. The + is the landing site

**Table. 1 Information on the CE-4 Landing Site**

| Element | Description |
|---|---|
| Landing site geographic coordinates | 177.5991°E, 45.4446°S, −5935 m. (The position uncertainty relative to CE2TMap2015 is meter level; See the Methods for details) |
| Coordinate system | The lunar coordinate is based on the mean Earth/polar axis coordinate system. The reference surface of the elevation is the surface of the Moon's spheroid with a radius of 1737.4 km, and the reference origin is the mass center of the Moon. |
| Landing site location in the images[a] | (1) In the CE2Tmap2015 DOM with a resolution of 7 m, landing site pixel coordinate (line 13602, column 21647) |
| | (2) In the LCAM image with a resolution of 5 m, landing site pixel coordinate (line 412, column 472) |
| | (3) In the LCAM image with a resolution of 1 m, landing site pixel coordinate (line 403, column 288) |
| | (4) In the LCAM image with a resolution of 10 cm, landing site pixel coordinate (line 592, column 389) |
| | (5) In the LCAM image with a resolution of 5 cm, landing site pixel coordinate (line 500, column 426) |

[a]The subdivision number for the CE2TMap2015 DOM and DEM with a resolution of 7 m is K136 in this study (Fig. 4a, b). The corresponding file names for LCAM 5 m, 1 m, 10 cm, and 5 cm resolution images are as follows:
CE4_LCAM_20190103022303_1740_08.tif (Fig. 5c),
CE4_LCAM_20190103022414_2444_08.tif (Fig. 5d),
CE4_LCAM_20190103022500_2906_08.tif (Fig. 5e),
and CE4_LCAM_20190103022525_3160_08.tif (Fig. 5f).

In contrast, the landing site of CE-4 is located southeast of the Von Kármán crater on the lunar farside. The length of the CE-4 nadir flight trace is also~450 km from north to south. However, the terrain of the nadir flight trace shows a large variation with a maximum variation of 6000 m (Fig. 6). Because the pointing position of the ranging beam is different from the final landing site and the terrain variation is obvious, if ranging measurements were introduced for a correction in advance, there would be a safety risk. Under its trace control strategy, CE-4 performed rapid attitude adjustments at an altitude of 8 ~ 6 km, and ranging measurements were introduced to correct the IMU bias and drift. This strategy ensured a safe landing of CE-4. The recovered trajectory showed that the lander attitude was adjusted to enable thrusting for near-vertical descent at an altitude of 5635 m. Thus, the powered descent trajectory of CE-4 is obviously different from that of CE3 (Fig. 7).

Based on the sequence images of the LCAM, we reconstructed the powered descent trajectory of the CE-4 lander through photogrammetry. Combining the generated cm-resolution terrain data near the landing site with the NCAM stereoscopic images, the geographical location of the landing site was calculated in this study. The horizontal and the vertical RMS errors of the landing site are 0.7 m (1δ) and 1.0 m (1δ) respectively (see the Methods section for details).

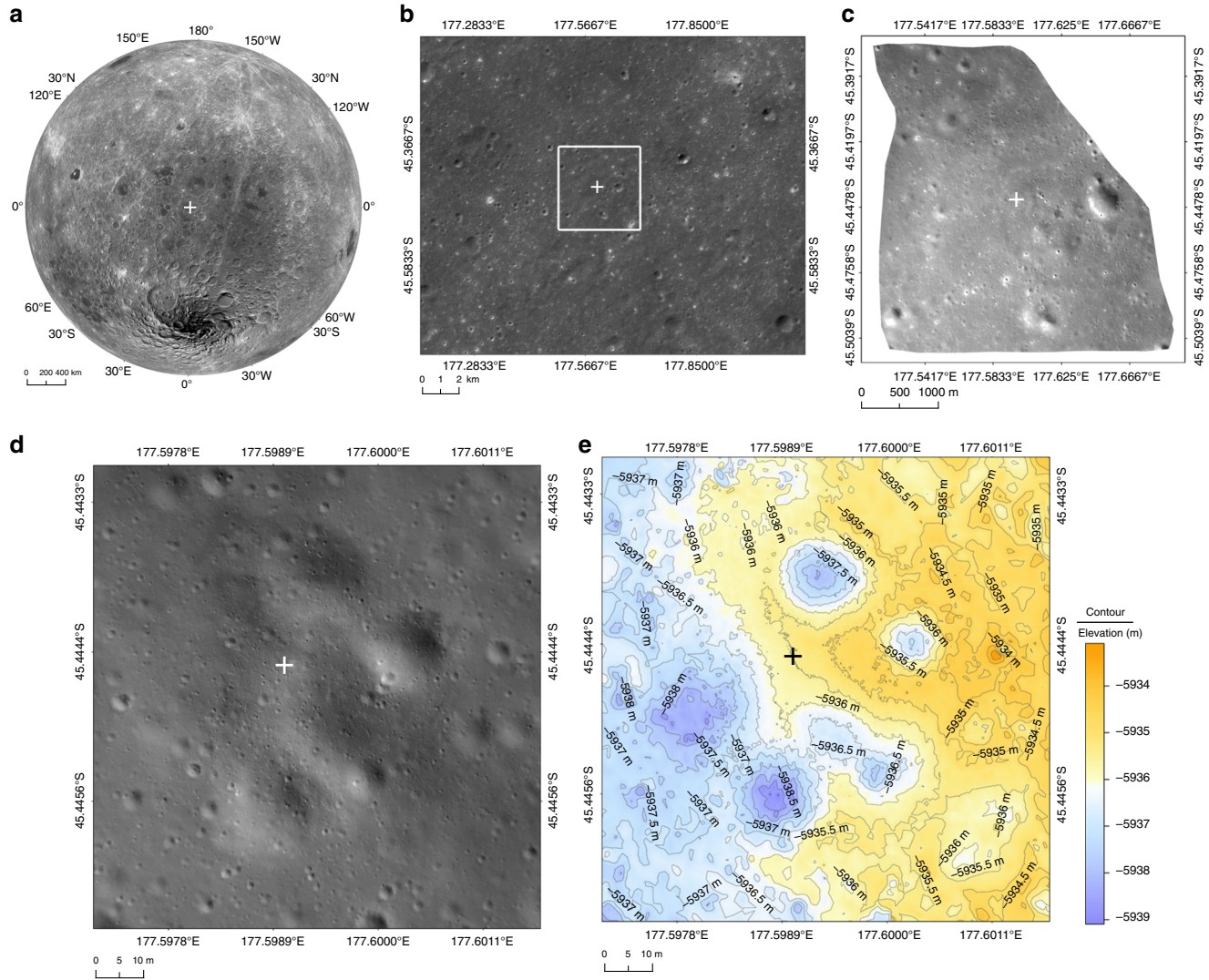

**Fig. 3** Location of the CE-4 landing site. The + is the identified landing location. **a**, **b** are the CE-2 DOMs, using the CE-4 landing site as the projection center of the azimuth projection. **c** is the LCAM DOM with a resolution of 5 m/p. **d** is the LCAM DOM, which is generated by the LCAM images with resolutions from 21.4 cm/p to 0.2 cm/p and is uniformly resampled at 5 cm/p during mapping. **e** is a shaded-relief map with contour line produced by the LCAM DEM, which area and resolution are the same as in (**d**)

Compared with the positioning results of the landing site based on LRO terrain data (177.5885°E, 45.4561°S, −5927m)[24], our results show a 226 m deviation along the latitude direction and a 348 m deviation along the longitude direction. The total positional deviation is 415 m, which reflects the deviation of the two sets of terrain data on the lunar farside (see the Methods section for details). A feasible method to effectively eliminate this deviation is to establish absolute control points on the lunar farside.

As a permanent artificial landmark on the lunar farside, the location of the CE-4 lander was precisely confirmed using CE-2 and CE-4 images and can serve as a potential control point on the farside. The result will provide a worthy geodetic data point for studies on lunar control points, high-precision lunar mapping, and subsequent lunar exploration, such as for the Yutu-2 rover.

## Methods

**Instruments and dataset descriptions**. The dataset used in this study includes landing camera (LCAM) images, navigation camera (NCAM) images and the Chang'E-2 (CE-2) global lunar terrain (CE2TMap2015)[20,21].

The LCAM is one of the scientific payloads installed on the bottom of the Chang'E-4 (CE-4) lander. According to the analysis of the characteristics of the

**Table 2 The performance parameters of the LCAM**

| No. | Name | Performance parameters |
|-----|------|------------------------|
| 1. | Wavelength range (nm) | 419~777 |
| 2. | Field of view (°) | 45.3° × 45.3° |
| 3. | Focal Length (mm) | 8.5 |
| 4. | Effective pixel numbers | 1024 × 1024 |
| 5. | Pixel size on Focal plane (μm) | 6.7 |
| 6. | Frame rate (fps) | 10 |
| 7. | Quantized value (bit) | 8 |
| 8. | Data compression ratio | 8:1 |

LCAM sequence images, the CE-4 lander gradually approached the landing area using a vertical descent mode after attitude adjusting phase. The overlap of the LCAM sequence images exceeds 94%. Over 40% of the images cover the landing site. The rotation and translation between adjacent images are slow. Therefore, 180 images in 1 s intervals during this phase were selected for the reconstruction of the powered descent trajectory and landing area topography. The main performance parameters of the LCAM are shown in Table 2.

The NCAM is an engineering payload onboard the CE-4 rover and is installed on the top of the rover mast. It consists of two optical systems. The NCAM obtained images surrounding the landing site at the top of the lander before the

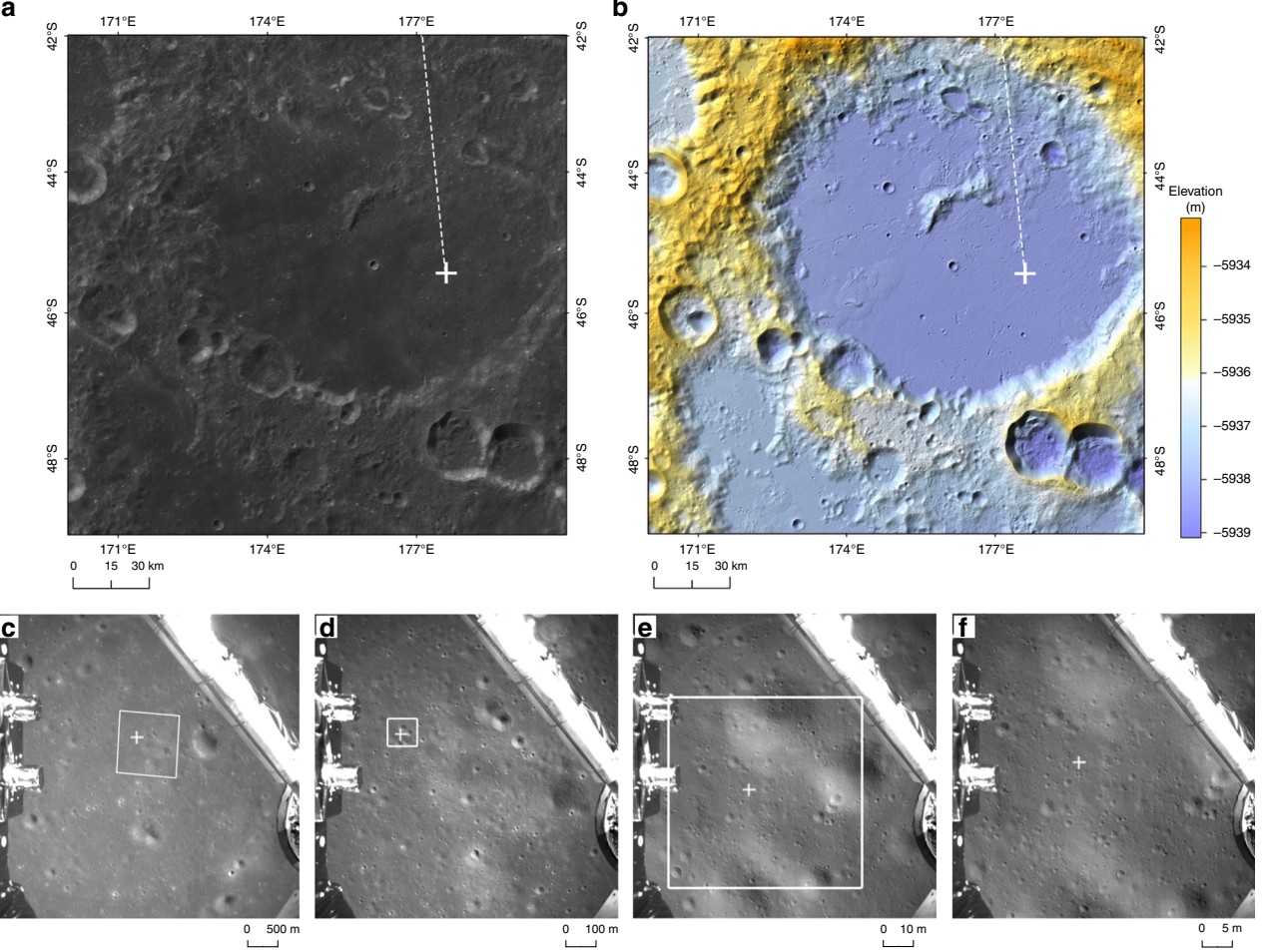

**Fig. 4** The CE-4 landing site on CE2TMap2015 and CE-4 LCAM images. The + is the identified landing location. **a** Landing site on the Chang'E-2 DOM. **b** Landing site on a shaded-relief map made by the Chang'E-2 DEM. The projection is the positive-axis isometric Mercator projection, which has a standard parallel at 45°S. White dotted line represents the approach trajectory of CE-4; **c–e**, f are the LCAM images with spatial resolutions of 5 m/p, 1 m/p, 10 cm/p, and 5 cm/p, respectively. The white rectangles in (**c–e**) represent the borders of (**d–f**), respectively

rover separated from the lander. Based on these images, topographic data of the landing site with cm-level accuracy can be reconstructed by close-range photogrammetry. Debris scattered near the landing site can be accurately identified. Because the relationship between the location of the NCAM and the CE-4 lander is known, the precise position of the landing site can be located by combining the images of the LCAM and NCAM[25]. The main performance parameters of the NCAM are shown in Table 3.

The georeferenced data used in this study are CE2TMap2015, which include the DOM and DEM with resolutions of 7 m/p and 20 m/p, respectively. These data are produced using CE-2 stereo images with a resolution of 7 m. The average horizontal and elevation relative position errors are 5 m and 2 m, respectively. Compared to the absolute positions of 5 laser reflectors located on the Moon, the horizontal positional deviation of CE2TMap2015 is 21 ~ 97 m, and the elevation deviation is 2 ~ 19 m[20,21].

LRO terrain data is used for CE-4 landing site position comparison. Orbit overlap analyses show that, LRO spacecraft ephemeris can be improved to less than 10 m horizontally and 1 m vertically by combining radiometric tracking data and LOLA data with the GRAIL gravity model[25–27]. As a result, the LRO terrain data uncertainty is about 20 meters[28] and the positional deviation from the 5 laser reflectors is <5.2 m[29]. A pair of LRO NAC observations of the CE-4 landing site were collected on 1 February 2019 (M1303619844LR, M1303640934LR), and were used to create a digital terrain model (DTM) on the LOLA-plus-GRAIL coordinate framework. The coordinates of CE-4 lander were derived (177.5885°E, 45.4561°S, −5927m)[24] based on the DTM.

It can be seen that both CE2TMap2015 and LRO terrain data show good internal consistency. However, the positioning deviation of the two sets of terrain data on the lunar farside will both increase due to several reasons, such as orbital measurement error, the lumpy gravity field of the Moon on the lunar farside, small uncertainties in camera model (distortion, FL), etc. Compared to the LRO terrain data, the average global positional deviation of CE2TMap2015 is 354 m with a standard deviation of 228 m (1δ)[20,21,30], which mainly results from large deviation

**Table 3 The performance parameters of the NCAM**

| No. | Name | Performance parameters |
|-----|------|------------------------|
| 1. | Focal Length (mm) | Left camera: 17.69 |
|    |      | Right camera: 17.98 |
| 2. | Effective pixel numbers | 1024 × 1024 |
| 3. | Pixel size on Focal plane (μm) | 15 |
| 4. | Baseline length (mm) | 269.69 |
| 5. | Quantized value (bit) | 8 |

of two sets of terrain data on the farside. The 415 m discrepancy between our coordinates and the LRO terrain data coordinates for the CE-4 lander is within the positional deviation range.

**Technical flowchart**. A total of 180 images (one image/per second) from the LCAM sequence images and 18 pairs of NCAM stereo images captured at the top of the CE4 lander were selected for this work. Figure 8 shows the technical flow-chart of the trajectory recovery and landing site position determination.

**CE-4 lander descent trajectory reconstruction**. First, evenly distributed points (or tie points) were automatically extracted using scale-invariant feature transform (SIFT) feature matching[31]. Then, least squares matching was applied to these tie points to achieve more precise matching results, and a random sample consensus (RANSAC)-based optimization algorithm was used to automatically eliminate mismatches[32,33]. A total of 30992 tie points, evenly distributed throughout the

**Fig. 5** Position of the CE-4 landing site. **a** The CE-4 landing site on the 5 cm/p resolution LCAM DOM, in which the + is the identified landing location and the # is the location of Yutu-2 on 12 January 2019. Five prominent craters around the landing site are marked. (**b**) CE-4 lander image obtained from the northwest of the landing site by the Yutu-2 PCAM at the location labeled in (**a**). (**c**) LRO NAC image of the CE-4 landing site collected on 1 February 2019 (M1303640934LR), in which the arrow indicates the location of CE-4 landing site. (**d**) Three-dimensional landscape map of the landing site generated by the DEM and DOM data through reconstruction of CE-4 LCAM sequential images acquired within 100 m above the lunar surface, in which the location of CE-4 lander and Yutu-2 is the same as labeled in (**a**). While landing position was determined from LCAM orbital data (**a**), the orientation of the lander was determined using PCAM image taken from the ground (**b**). Prominent craters are also marked in (**d**), as in (**a**)

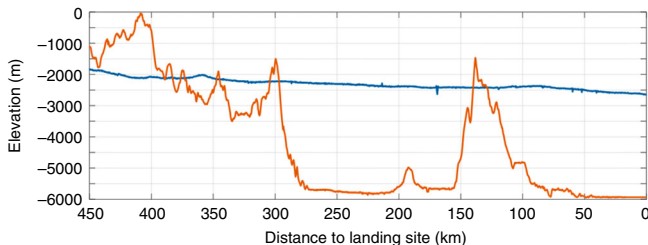

**Fig. 6** Elevation variations of the lunar terrain along the CE-3 and CE-4 nadir flight traces. The blue and orange lines represent the elevation variation for the CE-3 and CE-4 nadir flight trace, respectively. Elevation is given with respect to the Moon's spheroid with a radius of 1737.4 km

overlaps of the LCAM images, were extracted. The relative positional relationship of these images can be precisely restored according to these tie points.

Facilitated by one LRO NAC image (M178833263LC), 13 evenly distributed GCPs close to the landing site (including 12 horizontal and vertical GCPs and 1 vertical GCP) were extracted from the CE2TMap2015 map and LCAM images. The centers of some easily recognized small craters were mainly selected to ensure the selection accuracy of the GCPs.

According to photogrammetric bundle adjustment theory[22,23], the relationship between the lunar surface points, the corresponding image points and the camera projection center can be expressed by the classical collinear equation, which is as follows.

$$x - x_0 + \Delta x = -f \frac{r_{11}(X-X_s)+r_{21}(Y-Y_s)+r_{31}(Z-Z_s)}{r_{13}(X-X_s)+r_{23}(Y-Y_s)+r_{33}(Z-Z_s)}$$
$$y - y_0 + \Delta y = -f \frac{r_{12}(X-X_s)+r_{22}(Y-Y_s)+r_{32}(Z-Z_s)}{r_{13}(X-X_s)+r_{23}(Y-Y_s)+r_{33}(Z-Z_s)}$$

(1)

where $(x, y)$ are the image coordinates of the feature points, $(x0, y0)$ are the image coordinates of the image principal point, $(\Delta x, \Delta y)$ are self-calibration correction

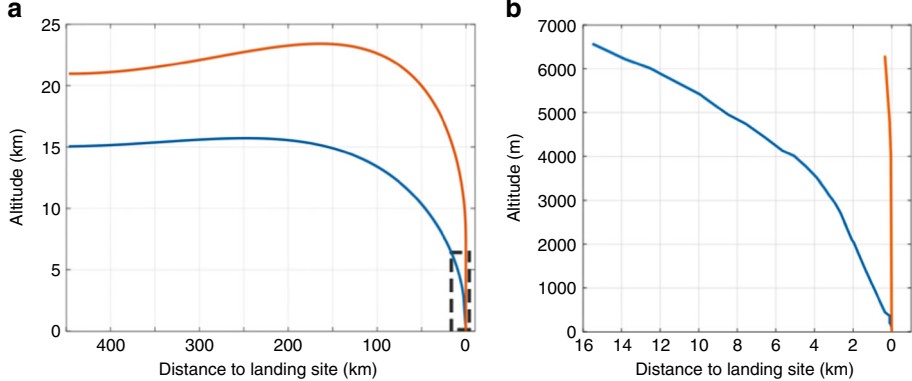

**Fig. 7** Comparison of powered descent trajectory between the CE-3 and CE-4. The blue and orange lines represent trajectories of the CE-3 and CE-4, respectively. **a** The designed trajectories of the entire powered descent, in which the marked rectangle corresponds to designed trajectories below the altitude of ~6 km; **b** Magnified portion of the reconstructed trajectory below the altitude of ~6 km. Altitude is given with respect to final landing site level

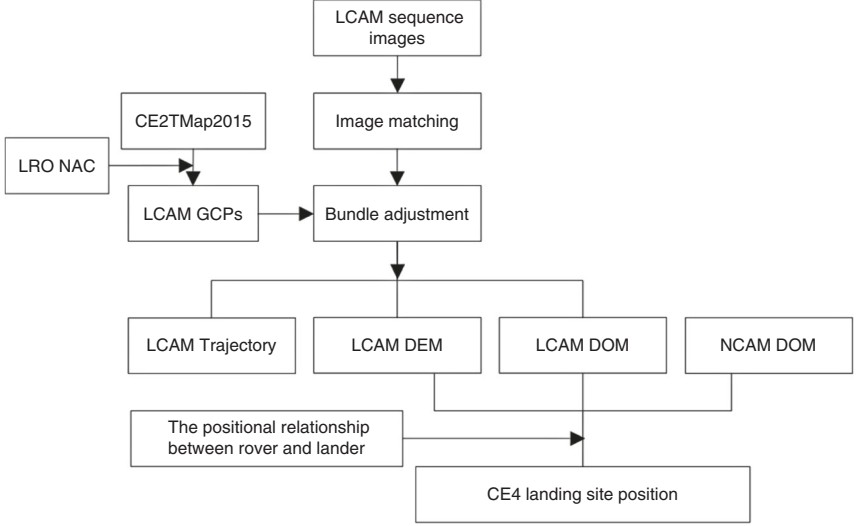

**Fig. 8** Data processing flowchart

terms of the camera, including lens distortion correction, $(X, Y, Z)$ are the lunar space coordinates of the feature points, $(X_s, Y_s, Z_s)$ are the lunar space coordinates of the image principal point, $f$ is the focal length of the camera, and $r11$–$r33$ are the elements of the rotation matrix $R(\varphi, \omega, \kappa)$, which is formed by the Euler attitude angle $(\varphi, \omega, \kappa)$ of the LCAM and used to translate feature points from the lunar space coordinates to the image coordinates. $(X_s, Y_s, Z_s, \varphi, \omega, \kappa)$ forms the exterior orientation (EO) parameters of the LCAM. Using Taylor's formula, the linearized error equations of each pixel can be derived by formula (1). The EO parameters of each image and the lunar space coordinates of each feature point can be solved by the least squares method. We implemented the photogrammetric bundle adjustment theory by ContextCapture Master software in this study. Therefore, we can use the location of each image to reconstruct the descent trajectory of the CE4 lander.

After the adjustment, the RMS error of the reprojection error for all the tie points is 0.5 pixels. The horizontal and vertical RMS errors of the GCPs are, respectively, 0.715 m (1δ) and 1.040 m (1δ) (Table 4), which represent the accuracy of the CE-4 descent trajectory reconstruction.

**Precise localization of the landing site**. First, the position and attitude of the LCAM images were obtained by the landing trajectory reconstruction. Then, a pixel-level point cloud was generated by the image dense matching technique[34,35], and a DEM of the LCAM with a 5 cm spatial resolution within 80 m around the landing site was interpolated from the triangulated irregular network (TIN). Finally, the same resolution and coverage DOM of the LCAM was produced using the image reprojection technique with the DEM and EO parameters.

Similarly, we produced the DEM and DOM of the NCAM with a 2 cm spatial resolution within 30 m around the landing site using the NCAM images. The only difference for this process is that the GCPs were selected from the LCAM terrain data. Considering that the spatial resolution of the NCAM image is close to that of the descent camera terrain data, some small stones scattered around the landing

**Table 4 Horizontal and vertical deviations of the GCPs for the LCAM**

| Name | RMS of reprojection error (pixel) | Horizontal error (m) | Vertical error (m) |
|---|---|---|---|
| LGCP01 | 0.35 | 1.522 | −2.319 |
| LGCP02 | 0.74 | 2.560 | 0.504 |
| LGCP03 | 0.21 | 1.327 | −1.528 |
| LGCP04 | 0.72 | 2.933 | 1.365 |
| LGCP05 | 0.55 | 0.897 | 0.807 |
| LGCP06 | 0.67 | 1.565 | −0.642 |
| LGCP07 | 0.72 | 1.685 | −0.513 |
| LGCP08 | 0.23 | 0.710 | 0.610 |
| LGCP09 | 0.93 | 0.795 | 0.799 |
| LGCP10 | 1.08 | 1.048 | −0.788 |
| LGCP11 | 0.75 | 0.855 | 0.420 |
| LGCP12 | 0.84 | 0.865 | −0.483 |
| LGCP13 | 0.11 | / | 0.065 |
| Global RMS | 0.61 | 0.715 | 1.040 |

site which can be easily identified were selected as GCPs to ensure the accuracy of the GCP selection (Fig. 9). A total of 14 GCPs were selected.

A total of 21,091 tie points were extracted using the image matching method. The RMS error of the reprojection error for all tie points is better than 0.2 pixels after the adjustment. The GCP horizontal RMS error is 0.017 m (1δ), and the vertical RMS

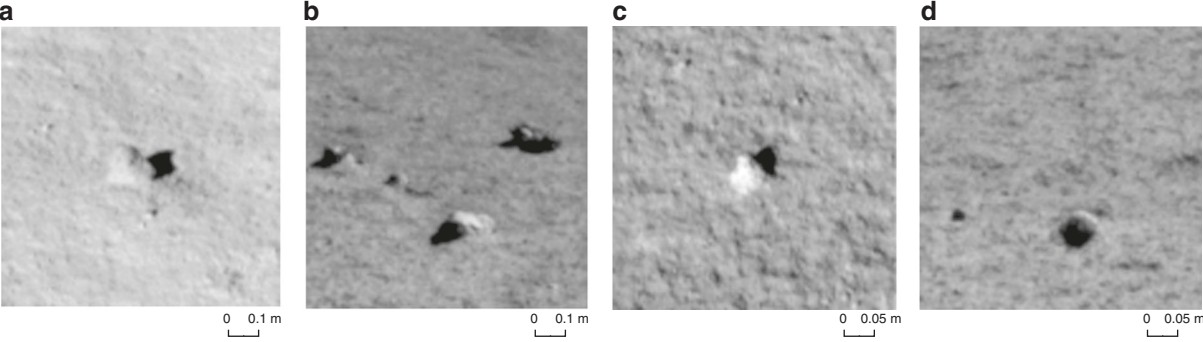

**Fig. 9** Small rocks used as GCPs. **a–d** are stones in the NCAM image that can also be easily identified in the LCAM DOM

**Table 5 Horizontal and vertical deviations of the GCPs for the NCAM**

| Name | RMS of the reprojection error (pixel) | Horizontal error (m) | Vertical error (m) |
|---|---|---|---|
| NGCP01 | 0.077 | 0.053 | −0.038 |
| NGCP02 | 0.006 | 0.034 | 0.038 |
| NGCP03 | 0.007 | 0.014 | −0.037 |
| NGCP04 | 0.010 | 0.032 | 0.048 |
| NGCP05 | 0.005 | 0.021 | 0.017 |
| NGCP06 | 0.007 | 0.028 | −0.065 |
| NGCP07 | 0.003 | 0.027 | −0.004 |
| NGCP08 | 0.007 | 0.040 | −0.009 |
| NGCP09 | 0.013 | 0.075 | 0.013 |
| NGCP10 | 0.005 | 0.049 | −0.027 |
| NGCP11 | 0.005 | 0.031 | 0.014 |
| NGCP12 | 0.005 | 0.020 | 0.001 |
| NGCP13 | 0.005 | 0.015 | 0.040 |
| NGCP14 | 0.017 | 0.014 | 0.015 |
| Global RMS | 0.012 | 0.017 | 0.031 |

error is 0.031 m (1δ) (Table 5). According to the positional relationship between the NCAM and the origin of the lander's body coordinate system (LBCS), the horizontal position of the vertical projection of the LBCS origin in the LCAM terrain data is 177.5991°E, 45.4446°S, and the elevation value is −5935 m.

## Data availability

CE2TMap2015 and CE-4 images are available at the Data Publishing and Information Service System of China's Lunar Exploration Program (http://moon.bao.ac.cn/). Datasets generated and/or analyzed during the current study are available from the corresponding author.

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

## Acknowledgements

This research was funded by the Chang'E-4 mission of Chinese Lunar Exploration Program (CLEP) and the National Natural Science Foundation of China (Grant No.11941002). We thank the team of the CE-4 Project for their successful work, especially the GRAS of CLEP for their valuable and efficient assistance with providing the data and data calibration.

## Author contributions

C.L.L., J.J.L., X.R., and W.Y. designed the research and wrote the paper. J.J.L, X.R., and W.Y. performed the calculations. Z.H. and J.Y. are the CE-4 Spacecraft System team members and helped with the NCAM data processing. X.G.Z., W.L.C., X.Y.G, D.W.L., X.T., X.X.Z., and T.N. helped with the LCAM, PCAM, and NCAM data calibration, processing and mapping. H.B.Z., W.Z. Y.S., and W.B.W are the Ground Research and Application System (GRAS) team members of and helped with data retrieval, data preprocessing and instrument operations.

## Additional information

**Competing interests:** The authors declare no competing interests.

