## [Peer Review File · Nature Communications]

Reviewers' comments:

Reviewer #1 (Remarks to the Author):

Review of Descent trajectory reconstruction and landing site positing of Chang'E-4 on Moon's farside

by

Jianjun Liu et al.,

General Comments

This paper describes the location of where the Chang'E-4 landed on the far side of the Moon and the technique used to identify the location. Traditional radiometric techniques are unusable because there is no line-of-sight with the far side, so there is no direct telemetry with Chang'E-4. As such, optical navigation techniques of triangulation based upon the Moon's digital terrain model were used, both in controlling the spacecraft for soft landing as well as computing where the spacecraft finally landed.

The paper is well written and clear with only a few awkward sentences.

The paper does a good job describing the sequence of events and the data that was available to the spacecraft and later the reconstruction team. The reviewer would have liked to see more detail in the method section of the specific mathematics and techniques used, especially the bundle adjustment applied.

In the discussion section: the comment about the gentle topography (L173) for the Chang'E-4 site compared to the Chang'E-3 site lacked sufficient detail to draw effective conclusions. A 3km variation vs a 6000m (please use the same units) does not seem like a drastic enough change. It seems like there was something significant, but the text was hard to follow. The phrase "avoid the discontinuity" (L176) needs more clarity.

L87 More information about the Automatic on-orbit analysis would be useful, either here or in methods. Is this software available? Does it have a name? Has it been used before?

L109 More information about the three-dimension laser imaging should be provided.

L139 Figure 4 shows a shaded relief map of the landing site. This shaded relief map has significant pixelation and artifacts (surface is too flat, lack of craters near the site, etc.) Can you provide any discussion for this? Also, it would be useful to have an image that shows which regions were done at what resolution. While the text says 2cm DEM were produced, I expect that wasn't done over all of the region. Thus, showing where the data has higher resolution would improve people's understanding of the process and where the data is best.

L146 How was the 5 laser reflectors integrated into your analysis? How where they measured? When where these measurements taken?

L233 More discussion about why LROC positions are so different than Chang'E-4 would be useful.

L245 Define RANSAC

L250 What is a vertical ground control point.

L256 Define EO

L270 Define TIN

L271 Define LACM, or did you mean LCAM?

Minor Comments

L14 successful should be successfully
L17 Recommend changing "real time" with "a" (It wasn't real time when the reconstruction was done)
L28-31 Sentence "After..." is awkward. Suggest replacing "experienced" with "conducted".
"descending" with "descent"
L31 Remove comma after Moon
L45 Define DOM and DEM
L49 Suggest replacing "Combined with the" with "Using combined"
L59 Caption. Suggest adding a comment that the + is the identified landing location
L70 Only use etc. for a list of items that are known. If you are making a list of items, give the full list. In this case, you might be able to remove etc altogether.
L78-82 Awkward sentence. This may be because the concept of vertical attitude isn't defined.
L81 Likewise "horizontal to the vertical direction" isn't clear
L89 Sentence starting with "Therefore..." is confusing.
L93 After hovered, replace comma with "and"
L113 Can you provide the speed of the "slow vertical descent?"
L117 Provide a reference for the bundle adjustment technique.
L123 Either provide the residuals for the bundle adjustment here or refer readers to the method section.
Table 1. Suggest putting error values next to the lat/lon rather than in comments
L157-159. I suggest putting a dashed line around the next image's footprint. It would make it easier for people to follow the progress of images. Thus, image C would have a small box that matched the full field of view of image D. Also, a scale bar on each would be beneficial.
L189-192 Suggest referring the reader to the methods section to understand where the RMS error comes from.
L195-196 Sentence starting with "Apparently, ..." is awkward.
L232 Where are the reflectors located? Are they on the near or far side? If so, how sampled?

Eric E. Palmer

Reviewer #2 (Remarks to the Author):

Review of Paper

„Descent Trajectory reconstruction and landing site positioning of Chang'E-4 on Moon's farside" by Jianjun Liu and Co-authors

This is a most interesting report on the first historic landing on the lunar far-side, quite appropriate for publication in Nature.

However, the text would benefit from improvements. While the spelling is good, the wording and phrasing need some more work effort. Many sentences are not sufficiently specific. Besides, the text leaves some open questions.

There is confusion about the instruments and the data that were used. Some critical information is only given late in the paper and should be moved up.

From the reading, I have the impression that in many places the paper is researching what the spacecraft was doing. I suggest to improve the phrasing to correct this impression.

I tried to point out problems and make some suggestions, see my comments below.

Generally speaking, I recommend publication of the exciting material, pending on a number of improvements of the text.

Specific comments:

Title:

..on Moon's farside -- > ..on the Moon's farside (or: on the lunar farside)

Abstract:

Because it is impossible to directly measure the farside of the Moon ... -- >

As Earth-based radio tracking is impossible for a spacecraft moving over the lunar farside ...

to use real-time series of image data -- >

to use image data obtained during descent, which were transmitted to the ground after landing via a relay satellite (perhaps you may want to give the name of the satellite and mention its important mission)

It was found that CE-4 ... realized a safe soft landing

-- > it seems that the paper is researching what the spacecraft was doing. I suggest to rephrase this (and in other places the main text of the paper.)

L 34 - 37

It is impossible to perform... real-time control based on the measurement and control network of earth -- > ? not clear, need to rephrase

L 40

It is expected that slight movement ... -- > We reconstructed the descent trajectory, showing even barely perceivable maneuvers of the spacecraft during the landing approach (ok?)

L 45

-- > please explain to the non-expert reader: what are DOM and DEM?

L 67

One general comment: It is not clear what is described in this chapter.

Is it the trajectory from the autonomous navigation or is it the reconstruction of the trajectory from the images, after the landing. The topic of trajectory reconstruction is only introduced later in the paper.

L 72

Autonomous navigation control was adopted in the whole landing process

-- > there is little information in the paper. Did the autonomous navigation use the LCAM images?

General question: Is the information on the autonomous navigation solution available, and was it used in the reconstruction of the trajectory?

L 73

obstacle avoidance -- > How was this accomplished?

L 78

... started acquiring image ...

-- > started acquiring images ... (or: acquired the first image...)

... started acquiring image 5m11s ... after onset of CE-4 power descent -- >

Can you give more information? What was the time of the touch-down?

(Perhaps, it is a good idea give a time line or a plot showing time, spacecraft speed and height?)

L 87

automatic onboard orbit analysis ... revealed ... surface slope -- > can you explain what was done?

L91

by optical imaging sensor on CE-4 -- >

Is this LCAM?

L 93

performed detailed detection and determined the landing site -- > performed detailed assessment of the morphology of the ground and a choice for a final landing site was made.

L 109

laser imaging sensor -- > can you explain?

L 115

General question: "We made LCAM topographic maps" -- > considering that the LCAM images were all taken from above, stereo angles between images were probably very small. How did you manage to produce the topography?

Were topographic maps produced after the trajectory reconstruction, assuming that the spacecraft position and attitude were fixed for each image? Or was this a combined approach?

Were the laser images available to support the map production?

L 121

Can you explain to the non-expert reader: what are GCPs?

What is the positioning accuracy of the CE-2 maps (from which GCPs were taken)?

(Information is given only later in the text)

L 134 Figure 3:

Please use consistently decimals of degree (preferred) -- or degree, minute, second!

Is the image on the right showing a detail within the image on the left?

It is confusing that the images right and left use identical numberings for different control points (if I am not mistaken?). Please choose better example to demonstrate control point work.

L 163 Figure 3 caption

I do not see the "red triangle"

Adjustment -- > adjustment

images Adjustment -- > image adjustment

L 167

Discussion -- > It is strange to have the discussion so early, even before description of Methods.

L 171

It is confusing to mix discussions of CE4 and CE3. What are we to learn from CE3 here?

L 177 fluctuating terrain -- > (?)

Figure 6: can you please mark the surface level?

The plot combines topography and spacecraft height. Would it make sense to show topography and spacecraft height separately?

L 174

our results reveal that the spacecraft switched from oblique downward to vertical downward...
-- > not clear (it seems that the paper is researching what the spacecraft was doing)

L 176

... which effectively avoid the discontinuity -- > discontinuity? not clear

... which effectively avoid -- > ... which effectively avoided

L 178

distance measuring sensor -- > can you explain?

General question: were the data from distance measuring sensor and the above mentioned laser imaging sensor used in the reconstruction of the trajectory?

(Note that these instruments are not listed in the "Instrument Description")

L 186

Based on the sequence images of the LCAM, we reconstructed the power descent trajectory of the CE-4 lander through the photogrammetry method

-- > this is coming very late (in the Discussion?). If I understand correctly, results are already shown earlier in the paper.

L 191 and L 232: CE2TMap2015 absolute accuracy...

-- > how did the map achieve this excellent accuracy? Unfortunately, the relevant papers that are referenced are all in Chinese.

L 195: Why is the difference in position so large? Does it mean that the LRO reference frame and CE2TMap2015 differ by ~500m? It seems there is some confusion about the reference frame in use.

L 206

Instrument descriptions -- > This information is coming very late in the paper. The complete list of instrument and relevant data used in the analysis should be briefly introduced earlier in the paper. The details may be moved to the end. It appears that critical instruments are not listed (laser imager, distance sensor?)

L 249: what is the NAC image used for? It is also not mentioned in Fig.7? Please clarify

L 369 We thank the team ... -- > perhaps this statement should go into the acknowledgements?

L 374 data ... available on reasonable request -- > "reasonable" ?

Reviewer #3 (Remarks to the Author):

The manuscript submitted by Liu et al presents the photogrammetric methodology utilized to determine the descent trajectory and surface coordinates of the Chang'e 4 lander.

1) What are the major claims of the paper?

- The authors derived a detailed reconstruction of the descent trajectory, which included autonomous hazard avoidance. They describe how the onboard systems avoided hazards and allowed the vehicle to land safely on the lunar farside. The trajectory and surface coordinate reconstructions were the result of standard photogrammetric bundle block adjustment techniques.

The authors provide the following coordinates for the lander 45.4446°S latitude, 177.5991°E, elevation -5935 meters and state that (lines 199-201) “As a permanent artificial landmark on the farside of the Moon, the location of the CE-4 lander was precisely confirmed using CE-2 and CE-4 image, and can serve as a potential control point on the farside of the Moon.”

Absolute coordinate control for the Chang'e 4 images was provided by tying the bundle block adjustment to the Chang'e 2 global image mosaic / control network, with a stated uncertainty of 21-97 m horizontal and 2-19 m elevation. The Chang'e 2 coordinate system is in turn tied to the 5 laser retroreflectors located on the nearside. Note that in the caption to Table 1 a footnote states (line 149): “...the elevation data was accurate to meter.” It is not clear to the reviewer where the accurate to a meter was derived when previously it was stated that the uncertainty in elevation was 2-19 m.

The authors compared their derived coordinates to those derived by the Lunar Reconnaissance Orbiter Camera (LROC) team values (45.457°S, 177.589°E, plus minus 20 meters) and state that there is a 376 m and 215 m difference in latitude and longitude (respectively), for a total 433 m positional offset. The authors state that: “The total positional deviation is 433 m. Apparently, the positioning data of this study is more sufficient and detailed, and the results are more accurate. The latitude and longitude disagreement with that of LRO data can be accounted for by different reference map.” Lines 195-198. [Note: a degree on the Moon is ~30.3 kilometer so 0.001° is ~30 meters]

2) Are they novel and will they be of interest to others in the community and the wider field?
- While the methods presented in this work to derive the trajectory and lander coordinates are not novel, the reconstruction of a robotic lander under autonomy as it sets down on the lunar farside is novel and will be of interest to others in the community and the wider field.

3) If the conclusions are not original, it would be helpful if you could provide relevant references. The conclusions seem to be original, though partially unsubstantiated. Authors might add a standard reference to the bundle block adjustment technique that goes back 60 years (see below).

4) Is the work convincing, and if not, what further evidence would be required to strengthen the conclusions?
- For the most part the work is convincing. I do have one quibble with the comparison to the LROC team coordinates of the lander summarized above (see also lines 194-198 in manuscript). One could justifiably argue that the CE-2 coordinate system tied to the 5 nearside retroreflectors will not be as accurate on the opposite side of the Moon for several reasons (i.e. the lumpy gravity field of the Moon, small uncertainties in camera model (distortion, FL) adding up over “distance”, etc.). For example, a control network derived from Clementine UVVIS images was also shown to have km offsets due to the aforementioned reasons (Speyerer et al., 2016).

The authors did not fully document the coordinate system used in their work (a key reference is in Chinese and this reviewer was unable to read the paper) thus the claim that discrepancy is due to “different reference map” is left ambiguous. The LROC coordinates derived have a stated accuracy much smaller than the 433 meters reported in this manuscript.

Lunar Reconnaissance Orbiter radiometric tracking data were combined with the GRAIL gravity model to significantly improve the Lunar Reconnaissance Orbiter spacecraft ephemeris and thus geodetic accuracy of each LOLA spot to less than 10 m horizontally and 1 m vertically (Lemoine et al. 2014; Mazarico et al. 2012; Mazarico et al. 2013). Subsequent, in-flight calibration has improved the mapping of a single NAC image to < 20 m (Speyerer et al., 2016). The methods used to control LROC NAC images to the LOLA framework (and associated uncertainties) are documented in Wagner et al (2017) and Henriksen et al (2017).

- Archinal, B.A., M.R. Rosiek, R.L. Kirk, and B.L. Redding, 2006. The Unified Lunar Control Network 2005. U.S. Geological Survey Open-File Report 2006-1367, 12 p.
- Henriksen, M. R, M. R. Manheim, K. N. Burns, P. Seymour, E. J. Speyerer, A. Deran, A. K. Boyd, E. Howington-Kraus, M. R. Rosiek, B. A. Archinal, and M. S. Robinson (2017), Extracting accurate and precise topography from LROC narrow angle camera stereo observations, *Icarus*, 283, 122-137, doi: 10.1016/j.icarus.2016.05.012.
- Lemoine, F.G., Goossens, S., Sabaka, T.J., et al., 2014. GRGM900C: A degree 900 lunar gravity model from GRAIL primary and extended mission data. *Geophys. Res. Lett.* 41, 3382–3389. doi: 10.1002/2014GL060027.
- Mazarico, E., Rowlands, D.D., Neumann, G.A., et al., 2012. Orbit determination of the lunar reconnaissance orbiter. *J. Geod.* 86 (3), 193–207. doi: 10.1007/s00190-011-0509-4.
- Mazarico, E., Goossens, S.J., Lemoine, F.G, et al. , 2013. Improved orbit determination of lunar orbiters with lunar and gravity fields obtained by the GRAIL mission. In: Proceedings of the 44th Lunar and Planetary Science Conference. The Woodlands, TX, Abstract #2414.
- Speyerer, E.J., Wagner, R.V., and Robinson, M.S., 2016. Geometric Calibration of the Clementine UVVIS Camera Using Images Acquired by the Lunar Reconnaissance Orbiter. In: The International Archives of the Photogrammetry, Remote Sensing and Spatial Information Sciences, Volume XLI-B4
- Speyerer, E.J., R.V. Wagner, M.S. Robinson, A. Licht, P.C. Thomas, K. Becker, J. Anderson, S.M. Brylow, D.C. Humm, M. Tschimmel, 2016. Pre-flight and On-orbit Geometric Calibration of the Lunar Reconnaissance Orbiter Camera, *Space Science Reviews*, 200, 357-392.
- Wagner, R. V., D. M. Nelson, J. B. Plescia, M. S. Robinson, E. J. Speyerer, and E. Mazarico (2017), Coordinates of anthropogenic features on the Moon, *Icarus*, 283, 92-103, doi: 10.1016/j.icarus.2016.05.011.

Yes, different reference maps were used: LOLA vs Chang'e 2 for the LROC vs Chang'e 4 work. The manuscript does not in any way document the assertion: "Apparently, the positioning data of this study is more sufficient and detailed, and the results are more accurate." The authors need to compare the Chang'e 2 coordinate system with the LOLA system in a global sense and make some quantitative assessment of any discrepancies between the two products to support the statement quoted above (lines 195-196 in the manuscript).

5) On a more subjective note, do you feel that the paper will influence thinking in the field? The description of the descent trajectory and derivation of the coordinates is certainly of broad interest, however I do not think the manuscript will influence thinking in the field.

General Comments To Authors

1) There are a few instances of inappropriate word usage and faulty grammar, likely due to translation to English. Some logical inconsistencies may wholly or partially be due to a language issue.

Examples:

1) Line 15 16: "Because it is impossible to directly measure the farside of the Moon, we cannot acquire the descent trajectory and the landing site of the spacecraft." It is possible to directly measure the lunar farside and the Chinese L2 relay satellite Queqiao should have been able to receive telemetry during the descent I cannot understand what is meant by this sentence. I can guess, but the authors should clarify. Perhaps the lander did not have the ability to transmit to Queqiao during the descent?

2) Please spell out terms before using abbreviations/acronyms (i.e. DOM, DEM)

3) "farside of the Moon" appears multiple times. After the first instance you can drop the "of the Moon" and just say farside.

4) "5m/p, 1m/p, 10cm/p, 5cm/p resolution" appears multiple times throughout the manuscript. Once or twice is plenty.

5) The values in Table 4 and 5 (and elsewhere in the manuscript) may be overly precise. Are three decimal places really justifiable (mm precision)?

Consider adding Classic Bundle Block Adjustment References:

Schmid, H.H., 1958, Eine allgemeine analytische Lösung für die Aufgabe der Photogrammetrie, Bildmessung und Luftbildwesen 1958, pp.103-113, 1959 pp.1-12. [German]

Brown, D. C., 1958. A Solution to the General Problem of Multiple Station Analytical Stereod-
angulation. Air Force Missile Test Center Report No. 58-8, Patrick AFB, Florida. [English]

First of all, we sincerely thank the reviewers for their critical reading of our manuscript, which help to significantly improve our manuscript. We addressed all remarks made by the 3 reviewers. Our responses are given in red below each remark.

Reviewers' comments:

Reviewer #1 (Remarks to the Author):

Review of Descent trajectory reconstruction and landing site positing of Chang'E-4 on Moon's farside by Jianjun Liu et al.,

General Comments

This paper describes the location of where the Chang'E-4 landed on the far side of the Moon and the technique used to identify the location. Traditional radiometric techniques are unusable because there is no line-of-sight with the far side, so there is no direct telemetry with Chang'E-4. As such, optical navigation techniques of triangulation based upon the Moon's digital terrain model were used, both in controlling the spacecraft for soft landing as well as computing where the spacecraft finally landed.

The paper is well written and clear with only a few awkward sentences.

The paper does a good job describing the sequence of events and the data that was available to the spacecraft and later the reconstruction team. The reviewer would have liked to see more detail in the method section of the specific mathematics and techniques used, especially the bundle adjustment applied.

Response :

According to the reviewer's opinion, we added a detailed description of the bundle adjustment theory in the "Methods" section. The original L253~L258 is revised as follows. "According to photogrammetric bundle adjustment theory^{22,23}, the relationship between the lunar surface points, the corresponding image points and the camera projection center can be expressed by the classical collinear equation, which is as follows.

$$\begin{aligned} x - x_0 + \Delta x &= -f \frac{r_{11}(X - X_s) + r_{21}(Y - Y_s) + r_{31}(Z - Z_s)}{r_{13}(X - X_s) + r_{23}(Y - Y_s) + r_{33}(Z - Z_s)} \\ y - y_0 + \Delta y &= -f \frac{r_{12}(X - X_s) + r_{22}(Y - Y_s) + r_{32}(Z - Z_s)}{r_{13}(X - X_s) + r_{23}(Y - Y_s) + r_{33}(Z - Z_s)} \end{aligned} \quad (1)$$

where (x, y) are the image coordinates of the feature points, (x_0, y_0) are the image coordinates of the image principal point, $(\Delta x, \Delta y)$ are self-calibration correction terms of the camera, including lens distortion correction, (X, Y, Z) are the lunar space coordinates of the feature points, (X_s, Y_s, Z_s) are the lunar space coordinates of the image principal point, f is the focal length of the camera, and $r_{11}-r_{33}$ are the elements of the rotation matrix $R(\varphi, \omega, \kappa)$, which is formed by the Euler attitude angle $(\varphi, \omega, \kappa)$ of the LCAM and is used to translate feature points from the lunar space coordinates to the image coordinates. $(X_s, Y_s, Z_s, \varphi, \omega, \kappa)$ forms the exterior orientation (EO) parameters of the LCAM. Using Taylor's formula, the linearized error equations of each pixel can be derived by formula (1). The EO parameters of each image and the lunar space coordinates of each feature point can be solved by the least squares method. We implemented photogrammetric bundle adjustment theory by software ContextCapture Master in this study. Therefore, we can use the location of each image to reconstruct the descent trajectory of the CE4 lander."

In the discussion section: the comment about the gentle topography (L173) for the Chang'E-4 site compared to the Chang'E-3 site lacked sufficient detail to draw effective conclusions. A 3km variation vs a 6000m (please use the same units) does not seem like a drastic enough change. It seems like there was something significant, but the text was hard to follow. The phrase "avoid the discontinuity" (L176) needs more clarity.

Response :

Thanks to the reviewer for his meticulous work. In order to draw our conclusions more clearly and effectively, the manuscript has been improved and supplemented accordingly. (1) In Figure 6, elevation variation of the lunar terrain along the CE-3 nadir flight traces is added, and the description is revised. The elevation variation of 3km in the original manuscript refers to the maximum elevation variation of the lunar terrain in the CE-3 pre-selected landing zone, rather than the actual ones along the CE-3 nadir flight traces. It can be seen from Fig. 6, the lunar terrain along the CE-3 nadir flight trace (about 450km) shows a flat, small variation, gradually decreasing from the south to the north with a maximum variation of ~800 m, while the terrain along the CE-4 nadir flight trace shows a large variation with a maximum variation of 6000 m. The difference in elevation variations of the lunar terrain along the CE-3 and CE-4 nadir flight traces is significant.

Fig. 6 Elevation variations of the lunar terrain along the CE-3 and CE-4 nadir flight traces. The blue and orange lines represent the elevation variation for the CE-3 and CE-4 nadir flight trace, respectively.

(2)The description in discussion section was revised. The characteristics of the powered descent trajectory of the CE-3 and CE-4 were compared and analyzed according to the difference of the on-orbit control strategies between the two missions. On the other hand, the comparison of powered descent trajectory between the CE-3 and CE-4 was also supplemented.

Fig. 7 Comparison of powered descent trajectory between the CE-3 and CE-4. The blue and orange lines represent trajectories of the CE-3 and CE-4, respectively. (a) The designed trajectories of the entire powered descent, in which the dotted border corresponds to designed trajectories below the altitude of ~6km; (b) The reconstruction trajectories below the altitude of ~6km.

(3) The unit of elevation variations in the manuscript has been changed to “meter”. The phrase “avoid the discontinuity” (L176) is deleted.

L87 More information about the Automatic on-orbit analysis would be useful, either here or in methods. Is this software available? Does it have a name? Has it been used before?

L109 More information about the three-dimension laser imaging should be provided.

Response :

During the process of CE-4 powered descent, sensors including laser ranging sensor, microwave ranging and velocimeter sensor, and an optical imaging sensor and laser imaging sensor were used for on-orbit real-time control. The telemetry data of these sensors was not completely transmitted to the ground and would be not released. Thus, the telemetry data was not used in this study.

Our work is focused on the descent trajectory reconstruction using high-frequency landing sequence images transmitted after safe landing. To avoid making people feel that the paper is researching what the spacecraft was doing, we have revised the manuscript in L89 and L109. The detailed process of the CE-4 on-orbit autonomous navigation and the sensors are no longer described, and only the characteristics of the powered descent trajectory reconstruction results are analyzed.

The Automatic on-orbit control technology and sensors including the laser imaging sensor of CE-4 is similar to Chang'E-3 (CE-3). We have added a brief introduction in the manuscript "Discussion" section with the following reference (Li et al., 2016).

Li, S., Jiang, X. Q., Tao, T. Guidance Summary and Assessment of the Chang'e-3 Powered Descent and Landing. JOURNAL OF SPACECRAFT AND ROCKETS. 53 (2): 258-277(2016)

L139 Figure 4 shows a shaded relief map of the landing site. This shaded relief map has significant pixelation and artifacts (surface is too flat, lack of craters near the site, etc.) Can you provide any discussion for this? Also, it would be useful to have an image that shows which regions were done at what resolution. While the text says 2cm DEM were produced, I expect that wasn't done over all of the region. Thus, showing where the data has higher resolution would improve people's understanding of the process and where the data is best.

Response :

According to the reviewer's opinion, we added a description of the spatial resolutions of the LCAM images used here. Figure 4 (e) is recreated and contour line is supplemented to improve people's understanding.

L139 Figure 4 (e) the DEM data is generated by the LCAM images with resolutions from 21.4 cm/p to 0.2 cm/p and is uniformly resampled at 5 cm/p during mapping. The CE-4 LCAM is a fixed focus camera with a focal length (8.55 mm). The camera obtained images during the dynamic process of the landing. Its image quality depends on distance and resolution, and the farther from the landing site, the lower the spatial resolution of the image.

L146 How was the 5 laser reflectors integrated into your analysis? How where they measured? When where these measurements taken?

Response :

The lunar position of these five laser reflectors is not directly used in the analysis of this paper, but is used to illustrate the absolute positional accuracy of CE2TMap2015. In addition to the reference (Li et al., 2018) in Chinese, two English references have been supplemented in this manuscript as supporting materials for CE2TMap2015 accuracy.

These 2 references are as follows:

*Yan, W., Liu, J., Ren, X., Wang, F., Wang, W., & Li, C. Orbit optimization of Chang'E-2 by global adjustment using images of the moon. *Advances in Space Research*.56(11):2389-2401(2015).*

*Ren, X., Liu, J.J., Li, C.L., et al. A Global Adjustment Method for Photogrammetric Processing of Chang'E-2 Stereo Images. *IEEE Transactions on Geoscience and Remote Sensing*, in press, (2019).*

Five laser reflectors (Apollo 11, Apollo 14, Apollo 15, Lunokhod 1 and Lunokhod 2) all placed on the near side of lunar surface. They are the highest measuring accuracy artificial marking points of the lunar surface until now. Using the lunar laser ranging retro-reflectors (LRRRs) technology, the measurement accuracy of the LRRRs position can be reached meter level (Williams et al., 1996, 2004; Murphy et al., 2011; Wagner et al., 2012).

The size of the laser reflector is less than 1 m, which cannot be recognized on current high-resolution lunar remote sensing images, while the lander or lunar rover can be identified on the image. In 2010, the recognition results of the five lunar modules on the LRO narrow-angle camera 0.5m resolution image is published on the NASA official website. According to the relative positional relationship between the laser reflector and the lunar module, (Williams et al., 1996 and 2004; Murphy et al., 2011; Wagner et al., 2012) calculated the lunar surface position of these lunar modules corresponding to the laser reflectors. The positioning accuracy is about 30m. We used CE-2 7m resolution image to register with the LRO narrow-angle camera 0.5m resolution image to determine the pixel position of the laser reflector on CE2TMap2015. Then the absolute positional accuracy of the moon surface of CE2TMap2015 is obtained (Yan et al, 2015; Li et al., 2018; Ren et al, 2019).

Related references are as follows.

Williams J. G., X X Newhall, and J. O. Dickey. "Lunar moments tides, orientation, and coordinate frames". *Planet. Space Sci.*, vol. 44, no. 10, pp. 1077-1080, (1996).

Williams J.G. Lunar Laser Ranging Science: Gravitational Physics and Lunar Interior and Geodesy. *35th COSPAR Scientific Assembly*, July 18-24, 2004, Paris, France, (2004)

Murphy T. W., E. G. Adelberger, J. B. R. Battat, et al. "Laser Ranging to the Lost Lunokhod~1 Reflector". *Icarus* vol. 211, pp. 1103-1108. (2011).

Wagner, R. V., Speyerer, E. J., Burns, K. N., Danton, J., Robinson, M. S. Revised Coordinates for Apollo Hardware. *ISPRS-International Archives of the Photogrammetry, Remote Sensing and Spatial Information Sciences*, 1, pp. 517-521. (2012).

L233 More discussion about why LROC positions are so different than Chang'E-4 would be useful.

Response :

The more discussion is added in the manuscript L233 and Methods.

The LROC positioning results are so different from the results in this study because there are differences in the absolute positions of the two sets of data products. Li et al., (2018) and Ren et al. (2019) compared the horizontal position deviation between CE2TMap2015 and LRO GLD100m. 839 checkpoints uniformly distributed over the entire lunar surface were selected. The planimetric deviations of the two data were calculated using the latitude and longitude coordinates of the checkpoints on the LRO GLD100m and CE2TMap2015. The average value was 354m with a standard deviation of 228 m (15). The horizontal position deviation of the CE-4 landing site determined by the two sets of data products is 433 m, which is reasonable within the positional deviation range between the CE2TMap2015 and LRO terrain data.

These 2 references are as follows:

*Yan, W., Liu, J., Ren, X., Wang, F., Wang, W., & Li, C. Orbit optimization of Chang'E-2 by global adjustment using images of the moon. *Advances in Space Research*.56(11):2389-2401(2015).*

*Ren, X., Liu, J.J., Li, C.L., et al. A Global Adjustment Method for Photogrammetric Processing of Chang'E-2 Stereo Images. *IEEE Transactions on Geoscience and Remote Sensing*, in press, (2019).*

L245 Define RANSAC

Response :

The full name of RANSAC is added in the manuscript, i.e. random sample consensus. RANSAC is a random sampling consistency algorithm, which is based on a set of sample data sets containing abnormal data, calculates mathematical model parameters of the data, and obtains effective sample data. In the image matching process, it is commonly used as an algorithm for erroneous matching point culling.

L250 What is a vertical ground control point.

Response :

Here, the sentence should be vertical control point.

The ground control points (GCPs) in the photogrammetric processing generally includes three categories: 1) horizontal control point; 2) vertical Control Point; 3) horizontal and vertical control point. The vertical Control Point is mainly used to control the value of the adjustment result in the elevation direction in the photogrammetric adjustment processing to ensure that the absolute accuracy of the adjustment result in the elevation direction is consistent with the elevation accuracy of the elevation control point.

L256 Define EO

Response :

The full name of EO is added in the manuscript, i.e. Exterior Orientation.

The EO parameters define the position and attitude information about each image in the object space during the photogrammetric adjustment process.

L270 Define TIN

Response :

The full name of TIN is added in the manuscript, i.e. Triangulated Irregular Network.

L271 Define LACM, or did you mean LCAM?

Response :

There is a spelling mistake. It should be a "LCAM" here.

Minor Comments

L14 successful should be successfully

L17 Recommend changing "real time" with "a" (It wasn't real time when the reconstruction was done)

L28-31 Sentence "After..." is awkward. Suggest replacing "experienced" with "conducted". "descending" with "descent"

L31 Remove comma after Moon

L49 Suggest replacing "Combined with the" with "Using combined"

L70 Only use etc. for a list of items that are known. If you are making a list of items, give the full list. In this case, you might be able to remove etc altogether.

L93 After hovered, replace comma with "and"

Response :

The manuscript has been revised according to the reviewer's comments.

L45 Define DOM and DEM

Response :

The full name of DOM and DEM is added in the manuscript, i.e. Digital Orthophoto Map and Digital Elevation Model.

L59 Caption. Suggest adding a comment that the + is the identified landing location

Response :

The interpretation of the "+" label, which represents the position of the CE-4 landing site, has been supplemented to the description of Figure 1(a).

L78-82 Awkward sentence. This may be because the concept of vertical attitude isn't defined.

L81 Likewise "horizontal to the vertical direction" isn't clear

Response :

This paragraph is mainly used to describe the changes of flight attitude of the CE-4 after rapid adjustment stage, that is, the flight attitude changed from the horizontal flight attitude to the vertical attitude to lunar surface. The manuscript is revised to avoid unclear description.

L89 Sentence starting with "Therefore..." is confusing.

Response :

This sentence has been deleted in the manuscript to avoid confusion.

L113 Can you provide the speed of the "slow vertical descent?"

Response :

During the “slow vertical descent” stage, the speed of the CE-4 decreased to approximately 1.5 m/s. The CE-4 speed change is supplemented in the manuscript during the process of describing the characteristics of the powered descent trajectory reconstruction results.

L117 Provide a reference for the bundle adjustment technique.

Response :

A reference for the bundle adjustment technique has been supplemented to the manuscript as follows:

Brown, D. C., 1958. A Solution to the General Problem of Multiple Station Analytical Stereogramulation. Air Force Missile Test Center Report No. 58-8, Patrick AFB, Florida.

L123 Either provide the residuals for the bundle adjustment here or refer readers to the method section.

Response :

The bundle adjustment technique has been used in two parts in the "Methods" section of this manuscript, that is, 1) In “CE-4 Lander Descent Trajectory reconstruction” section. After the adjustment, the image point residuals are statistically analyzed. The original L259~261 position gives the analysis results. It is 0.5 pixels; 2) In “precise localization of landing site” section, statistical analysis of the image point residual after adjustment, the original L285~287 position gives the analysis result, which is 0.2 pixels.

Table 1. Suggest putting error values next to the lat/lon rather than in comments

Response :

The accuracy of control points in Table 4 reflects the accuracy of the landing site positioning. According to the reviewer's opinion, it is placed next to the landing site positioning result of Table 1. The original manuscript is revised to “177.5991°E ,45.4446°S , -5935 m (The position uncertainty relative to CE2TMap2015 is meter level. The absolute position uncertainty is several decameters; see the Methods for details)”.

L157-159. I suggest putting a dashed line around the next image's footprint. It would make it easier for people to follow the progress of images. Thus, image C would have a small box that matched the full field of view of image D. Also, a scale bar on each would be beneficial.

Response :

Figure 5(c)~ (d) are recreated as suggested. The progress in the sequence images and a scale bar to each image are supplemented.

L189-192 Suggest referring the reader to the methods section to understand where the RMS error comes from.

Response :

A sentence of "see Methods section for details" is added here in the manuscript to refer the reader to the "Methods" section to understand where the RMS error comes from.

L195-196 Sentence starting with "Apparently, ..." is awkward.

Response :

Through careful discussion, we revised the conclusion of this sentence to " The position of the CE-4 landing site determined in this study is authentic, which can reveal the actual location on the lunar farside. In this study, the CE-4 descent trajectory was recovered, and the landing site position was accurately determined. The pixel coordinates of the landing site were given in multiple LCAM images."

L232 Where are the reflectors located? Are they on the near or far side? If so, how sampled?

Response :

Five laser reflectors (Apollo 11, Apollo 14, Apollo 15, Lunokhod 1 and Lunokhod 2) all placed on the near side of lunar surface. They are the highest measuring accuracy artificial marking points of the lunar surface until now. Using the lunar laser ranging retro-reflectors (LRRRs) technology, the measurement accuracy of the LRRRs position can be reached meter level (Williams et al., 1996, 2004; Murphy et al., 2011; Wagner et al., 2012).

The size of the laser reflector is less than 1 m, which cannot be recognized on current high-resolution lunar remote sensing images, while the lander or lunar rover can be identified on the image. In 2010, the recognition results of the five lunar modules on the LRO narrow-angle camera 0.5m resolution image is published on the NASA official website. According to the relative positional relationship between the laser reflector and the lunar module, (Williams et al., 1996 and 2004; Murphy et al., 2011; Wagner et al., 2012) calculated the lunar surface position of these lunar modules corresponding to the

laser reflectors. The positioning accuracy is about 30m. We used CE-2 7m resolution image to register with the LRO narrow-angle camera 0.5m resolution image to determine the pixel position of the laser reflector on CE2TMap2015. Then the absolute positional accuracy of the moon surface of CE2TMap2015 is obtained (Yan et al, 2015; Li et al., 2018; Ren et al, 2019).

Related references are as follows:

Yan, W., Liu, J., Ren, X., Wang, F., Wang, W., & Li, C. Orbit optimization of Chang'E-2 by global adjustment using images of the moon. *Advances in Space Research*.56(11):2389-2401(2015).

Li, C. L., Liu, J. J., Ren, X., et al. Lunar Global High precision Terrain Reconstruction Based on Chang'e-2 Stereo Images. *Geomatics and information Science of Wuhan University*. 43 (4): 486-495 (2018).

Ren, X., Liu, J.J., Li, C.L., et al. A Global Adjustment Method for Photogrammetric Processing of Chang'E-2 Stereo Images. *IEEE Transactions on Geoscience and Remote Sensing*, in press, (2019).

Williams J. G., X X Newhall, and J. O. Dickey. "Lunar moments tides, orientation, and coordinate frames". *Planet. Space Sci.*, vol. 44, no. 10, pp. 1077-1080, (1996).

Williams J.G. Lunar Laser Ranging Science: Gravitational Physics and Lunar Interior and Geodesy. 35th COSPAR Scientific Assembly, July 18-24, 2004, Paris, France, (2004).

Murphy T. W., E. G. Adelberger, J. B. R. Battat, et al. "Laser Ranging to the Lost Lunokhod~1 Reflector". *Icarus* vol. 211, pp. 1103-1108. (2011).

Wagner, R. V., Speyerer, E. J., Burns, K. N., Danton, J., Robinson, M. S. Revised Coordinates for Apollo Hardware. *ISPRS-International Archives of the Photogrammetry, Remote Sensing and Spatial Information Sciences*, 1, pp. 517-521. (2012).

Eric E. Palmer

Reviewer #2 (Remarks to the Author):

Review of Paper

„Descent Trajectory reconstruction and landing site positioning of Chang'E-4 on Moon's farside“ by Jianjun Liu and Co-authors

This is a most interesting report on the first historic landing on the lunar far-side, quite appropriate for publication in Nature.

However, the text would benefit from improvements. While the spelling is good, the wording and phrasing need some more work effort. Many sentences are not sufficiently specific. Besides, the text leaves some open questions.

There is confusion about the instruments and the data that were used. Some critical information is only given late in the paper and should be moved up.

Response :

This work is focused on the descent trajectory reconstruction using high-frequency landing sequence images transmitted by the relay satellite after safe landing, and landing site positioning of CE-4 using combined binocular stereoscopic images obtained by the navigation camera (NCAM) and lunar global topographic data CE2TMap2015. The data of the on-orbit autonomous navigation system sensors are not used here. The detailed process of the CE-4 on-orbit autonomous navigation and the sensors are no longer described, and only the characteristics of the powered descent trajectory reconstruction results are analyzed.

The Automatic on-orbit control technology and sensors including the laser imaging sensor of CE-4 is similar to Chang'E-3 (CE-3). We have added a brief introduction in the manuscript "Discussion" section with the following reference (Li et al., 2016).

Li, S., Jiang, X. Q., Tao, T. Guidance Summary and Assessment of the Chang'e-3 Powered Descent and Landing. JOURNAL OF SPACECRAFT AND ROCKETS. 53 (2): 258-277(2016)

According to the format requirements of the Nature communication journal, the instruments and data used in this study are introduced in the "Methods" section after the Main text (including Introduction, results, discussion).

From the reading, I have the impression that in many places the paper is researching what the spacecraft was doing. I suggest to improve the phrasing to correct this impression.

Response :

The main purpose of this study is the descent trajectory reconstruction and the precise landing site positioning of CE-4. We have revised the textual description of the manuscript to focus on the trajectory characteristics of the powered descent stage of the CE-4.

I tried to point out problems and make some suggestions, see my comments below.

Generally speaking, I recommend publication of the exciting material, pending on a number of improvements of the text.

Specific comments:

Title:

..on Moon's farside -- > ..on the Moon's farside (or: on the lunar farside)

Abstract:

Because it is impossible to directly measure the farside of the Moon ... -- >

As Earth-based radio tracking is impossible for a spacecraft moving over the lunar farside ...

to use real-time series of image data -- >

to use image data obtained during descent, which were transmitted to the ground after landing via a relay satellite (perhaps you may want to give the name of the satellite and mention its important mission)

重要使命)

Response :

The manuscript has been revised according to the reviewer's comments.

It was found that CE-4 ... realized a safe soft landing

-- > it seems that the paper is researching what the spacecraft was doing. I suggest to rephrase this (and in other places the main text of the paper.)

Response :

This sentence has been deleted in the abstract to avoid making people feel that the paper is researching what the spacecraft was doing.

L 34 – 37

It is impossible to perform... real-time control based on the measurement and control network of earth -- > ?
not clear, need to rephrase

Response :

The manuscript has been revised to make the meaning of the expression clearer. The revised content is as follows:

"Because there was no radio measurement equipment between the CE-4 and the relay satellite Queqiao, it was unable to perform direct or indirect radio measurements by the ground tracking network on the lunar farside. In addition, the telemetry data received by Queqiao (including the detector altitude, acceleration, and attitude) were not released and

couldn't to be used. As a result, it is difficult to accurately reconstruct the spacecraft's landing trajectory and to confirm the precise landing site location. However, these problems can be effectively solved through localization technology based on landing images, which is not affected by factors such as the lunar gravity field and the dynamical model.”

L 40

It is expected that slight movement ...  We reconstructed the descent trajectory, showing even barely perceivable maneuvers of the spacecraft during the landing approach (ok?)

Response :

The manuscript has been revised according to the reviewer's comments.

L 45

 please explain to the non-expert reader: what are DOM and DEM?

Response :

The full name of DOM and DEM is added in the manuscript, i.e. Digital Orthophoto Map and Digital Elevation Model.

L 67

One general comment: It is not clear what is described in this chapter.

Is it the trajectory from the autonomous navigation or is it the reconstruction of the trajectory from the images, after the landing. The topic of trajectory reconstruction is only introduced later in the paper.

Response :

The main purpose of this study is to reconstruct CE-4 powered descent trajectory using high-frequency landing sequence images transmitted by the relay satellite after safe landing and no data of on-orbit autonomous navigation system sensors is used. The detailed process of the CE-4 on-orbit autonomous navigation and the sensors are no longer described, and only the characteristics of the powered descent trajectory reconstruction results are analyzed.

According to the format requirements of the Nature communication journal, detailed method of trajectory reconstruction used in this study are introduced in the “Methods” section after the Main text (including Introduction, results, discussion).

L 72

Autonomous navigation control was adopted in the whole landing process

-- > there is little information in the paper. Did the autonomous navigation use the LCAM images?

General question: Is the information on the autonomous navigation solution available, and was it used in the reconstruction of the trajectory?

Response :

The LCAM images were not used in the autonomous navigation process, but they completely recorded the entire powered descent process of the CE-4. This is why we used the LCAM images to reconstruct the powered descent trajectory.

During the process of CE-4 powered descent, sensors including laser ranging sensor, microwave ranging and velocimeter sensor, and an optical imaging sensor and laser imaging sensor were used for on-orbit real-time control. The telemetry data of these sensors was not completely transmitted to the ground and would be not released. Thus, the telemetry data was not used in this study.

Our work is focused on the descent trajectory reconstruction using high-frequency landing sequence images transmitted after safe landing. To avoid making people feel that the paper is researching what the spacecraft was doing, we have revised the manuscript. The detailed process of the CE-4 on-orbit autonomous navigation and the sensors are no longer described, and only the characteristics of the powered descent trajectory reconstruction results are analyzed.

The Automatic on-orbit control technology and sensors including the laser imaging sensor of CE-4 is similar to Chang'E-3 (CE-3). We have added a brief introduction in the manuscript "Discussion" section with the following reference (Li et al., 2016).

Li, S., Jiang, X. Q., Tao, T. Guidance Summary and Assessment of the Chang'e-3 Powered Descent and Landing. JOURNAL OF SPACECRAFT AND ROCKETS. 53 (2): 258-277(2016)

L 73

obstacle avoidance -- > How was this accomplished?

Response :

We added a brief introduction of the CE-4 on-orbit autonomous navigation control process and the sensors of navigation system in the "Discussion" section with a reference as follows.

Li, S., Jiang, X. Q., Tao, T. Guidance Summary and Assessment of the Chang'e-3 Powered Descent and Landing. *JOURNAL OF SPACECRAFT AND ROCKETS*. 53 (2): 258-277(2016)
Because the Automatic on-orbit control technology and navigation system sensors of CE-4 are similar to Chang'E-3 (CE-3), the specific implementation process of the obstacle avoidance process also can be found in the above reference.

L 78

... started acquiring image ...

-- > started acquiring images ... (or: acquired the first image...)

Response :

The manuscript has been revised according to the reviewer's comments.

... started acquiring image 5m11s ... after onset of CE-4 power descent -- >

Can you give more information? What was the time of the touch-down?

(Perhaps, it is a good idea give a time line or a plot showing time, spacecraft speed and height?)

Response :

It is confirmed that the LCAM began to obtain images 5 m35s after the CE-4 entering the powered descent stage, which lasted 11 m27s and then the CE-4 landed safely. In this manuscript, the description of the CE-4 speed change is supplemented during the process of describing the characteristics of the powered descent trajectory reconstruction results, while an comparison of powered descent trajectory between the CE-3 and CE-4.

L 87

automatic onboard orbit analysis ... revealed ... surface slope -- > can you explain what was done?

Response :

The specific implementation process for on-orbit autonomous navigation control can be found in the following references. The detailed process of the CE-4 on-orbit autonomous navigation is no longer described in this manuscript, and only the characteristics of the powered descent trajectory reconstruction results are analyzed.

Li, S., Jiang, X. Q., Tao, T. Guidance Summary and Assessment of the Chang'e-3 Powered Descent and Landing. *JOURNAL OF SPACECRAFT AND ROCKETS*. 53 (2): 258-277(2016)

L91

by optical imaging sensor on CE-4 -- >

Is this LCAM?

Response :

The optical imager here does not indicate the LCAM, but one of the on-orbit autonomous navigation system sensors of the CE-4. We added a brief introduction in the "Discussion" section with a reference as follows.

Li, S., Jiang, X. Q., Tao, T. Guidance Summary and Assessment of the Chang'e-3 Powered Descent and Landing. *JOURNAL OF SPACECRAFT AND ROCKETS*. 53 (2): 258-277(2016)

L 93

performed detailed detection and determined the landing site -- > performed detailed assessment of the morphology of the ground and a choice for a final landing site was made.

Response :

The detailed process of the CE-4 on-orbit autonomous navigation is no longer described in this manuscript, and only the characteristics of the powered descent trajectory reconstruction results are analyzed. This sentence has been deleted in the manuscript.

L 109

laser imaging sensor -- > can you explain?

Response :

During the process of CE-4 powered descent, sensors including laser ranging sensor, microwave ranging and velocimeter sensor, and an optical imaging sensor and laser imaging sensor were used for on-orbit real-time control. The telemetry data of these sensors was not completely transmitted to the ground and would be not released. Thus, the telemetry data was not used in this study.

Our work is focused on the descent trajectory reconstruction using high-frequency landing sequence images transmitted after safe landing. To avoid making people feel that the paper is researching what the spacecraft was doing, we have revised the manuscript in L109. The detailed process of the CE-4 on-orbit autonomous navigation and the sensors are no longer described, and only the characteristics of the powered descent trajectory reconstruction results are analyzed.

The Automatic on-orbit control technology and sensors including the laser imaging sensor of CE-4 is similar to Chang'E-3 (CE-3). We have added a brief introduction in the manuscript "Discussion" section with the following reference (Li et al., 2016).

Li, S., Jiang, X. Q., Tao, T. Guidance Summary and Assessment of the Chang'e-3 Powered Descent and Landing. JOURNAL OF SPACECRAFT AND ROCKETS. 53 (2): 258-277(2016)

L 115

General question: "We made LCAM topographic maps"  considering that the LCAM images were all taken from above, stereo angles between images were probably very small. How did you manage to produce the topography?

Response :

In traditional aerial photogrammetry, stereo images are obtained by the multi-angle observations of the same target. These images need to meet certain overlap requirements. The corresponding stereo angles between images can construct stable observation geometry (i.e. stereo observation geometry). So the three-dimensional coordinates of the target can be solved using the geometry. In principle, if any two consecutive descent images are strictly on the same vertical line, there is no parallax at the center of the images. During the powered descent of the CE-4 lander, there are small deviations between the centers of the adjacent images. There are also small deviations between the observation directions of the adjacent images because of the different spatial resolutions. The stereo angles between adjacent images can be formed in the two cases. In order to obtain better stereo observation geometry, 180 images with 1 s intervals from the sequence LCAM images were selected. Now, the stereo observation geometry is better than that of the original sequence LCAM images. In lines 146-149 in the manuscript, the results of the LCAM adjustment show that this way was very effective.

The position and attitude of the LCAM images were obtained after the adjustment. A pixel-level point cloud was generated by the image dense matching technique, and a DEM of the LCAM with a 5 cm spatial resolution within 80 m around the landing site was interpolated from the triangulated irregular network (TIN).

In order to understanding, we added the following reference in the "Methods" section.

Li, R., Ma, F., Xu, F., et al. Localization of Mars rovers using descent and surface - based image data. Journal of Geophysical Research: Planets, 107(E11), 8004 (2002).

Were topographic maps produced after the trajectory reconstruction, assuming that the spacecraft position and attitude were fixed for each image? Or was this a combined approach?

Response :

The topographic maps were produced after the trajectory reconstruction. After the adjustment processing, the position and attitude information of each LCAM image at the imaging time can be obtained. These data will be changed with the landing trajectory so that they can be used for the reconstruction of powered descent trajectory. The detailed description of bundle adjustment method is implemented in the “CE-4 lander descent trajectory reconstruction” section of “Methods” section. When the position and attitude of the LCAM images were obtained by the landing trajectory reconstruction, a pixel-level point cloud was generated by the image dense matching technique and a DEM was interpolated from the triangulated irregular network (TIN). The detailed description is also implemented in the “Methods” section.

Were the laser images available to support the map production?

Response :

During the process of CE-4 powered descent, sensors including laser ranging sensor, microwave ranging and velocimeter sensor, and an optical imaging sensor and laser imaging sensor were used for on-orbit real-time control. The telemetry data of these sensors was not completely transmitted to the ground and would be not released. Thus, the telemetry data was not used in this study.

Our work is focused on the descent trajectory reconstruction using high-frequency landing sequence images transmitted after safe landing. To avoid making people feel that the paper is researching what the spacecraft was doing, we have revised the manuscript. The detailed process of the CE-4 on-orbit autonomous navigation and the sensors are no longer described, and only the characteristics of the powered descent trajectory reconstruction results are analyzed.

The Automatic on-orbit control technology and sensors including the laser imaging sensor of CE-4 is similar to Chang'E-3 (CE-3). We have added a brief introduction in the manuscript “Discussion” section with the following reference (Li et al., 2016).

Li, S., Jiang, X. Q., Tao, T. Guidance Summary and Assessment of the Chang'e-3 Powered Descent and Landing. JOURNAL OF SPACECRAFT AND ROCKETS. 53 (2): 258-277(2016)

L 121

Can you explain to the non-expert reader: what are GCPs?

What is the positioning accuracy of the CE-2 maps (from which GCPs were taken)?

(Information is given only later in the text)

Response :

A control point is a point that has a ground fixed mark and coordinates (measured within a certain accuracy range) and has a function of location datum, including horizontal control point, vertical/elevation Control Point and horizontal/ vertical control point. The control point information used for the LCAM images and the NCAM images adjustment processing is described in L249~L252 , L274~L278 of the manuscript. In this study, the control point lunar coordinates which used in the LCAM images photogrammetry processing are measured from the CE2TMap2015.

The average horizontal and elevation relative position deviations of the CE2TMap2015 are 5 m and 2 m, respectively. Compared to the absolute positions of 5 laser reflectors located on the lunar surface, the horizontal positional deviation of CE2TMap2015 is 21~97 m, and the elevation deviation is 2~19 m. (Yan et al.,2015; Li et al., 2018 ,Ren et al., 2019)

References:

Yan, W., Liu, J., Ren, X., Wang, F., Wang, W., & Li, C. Orbit optimization of Chang'E-2 by global adjustment using images of the moon. *Advances in Space Research*.56(11):2389-2401(2015).

Li, C. L., Liu, J. J., Ren, X., et al. Lunar Global High precision Terrain Reconstruction Based on Chang'e-2 Stereo Images. *Geomatics and information Science of Wuhan University*. 43 (4): 486-495 (2018).

Ren, X., Liu, J.J., Li, C.L., et al. A Global Adjustment Method for Photogrammetric Processing of Chang'E-2 Stereo Images. *IEEE Transactions on Geoscience and Remote Sensing*, in press, (2019).

L 134 Figure 3:

Please use consistently decimals of degree (preferred) -- or degree, minute, second!

Is the image on the right showing a detail within the image on the left?

It is confusing that the images right and left use identical numberings for different control points (if I am not mistaken?). Please choose better example to demonstrate control point work.

Response :

The latitude and longitude values in the Figure 3 are in degrees. The image on the right does not showing a detail within the image on the left. Figure 3 (a) is a the spatial distribution of control points for the LCAM image adjustment, which is selected on CE2TMap2015 DOM data, while Figure 3 (b) is the spatial distribution of control points for the NCAM image adjustment, which is selected on DOM produced by LCAM images. The control point number in the figure is revised here. The number in the Figure 3 (a) is changed to "LGCP01, LGCP02... LGCP13", and the number in Table 4 is also revised. On the other hand, the number in the Figure 3 (b) is changed to "NGCP01, NGCP02... NGCP14", and the number in Table 5 is also revised.

L 163 Figure 3 caption

I do not see the "red triangle"

Response :

It should be white cross here. The manuscript has been revised according to the reviewer's comments.

Adjustment -- > adjustment

images Adjustment -- > image adjustment

Response :

The manuscript has been revised according to the reviewer's comments.

L 167

Discussion -- > It is strange to have the discussion so early, even before description of Methods.

Response :

According to the format requirements of the Nature communication journal, the "Methods" section should be described after the conclusion and discussion section as supporting materials.

L 171

It is confusing to mix discussions of CE4 and CE3. What are we to learn from CE3 here?

Response :

In order to eliminate confusion, the details description has been revised in the “Discussion” section. The difference in the powered descent trajectories of CE-3 and CE-4 is mainly discussed.

We compared the trajectory and nadir trace elevation variations of CE-3 with that of CE-4, and found that the change of the CE-4 orbital strategy mainly results from the terrain variation. This strategy shows a good performance in the CE-4 mission. Thus, consideration of the terrain variation when formulating the orbital strategy is important for future safe soft landing on the planet surface.

L 177 fluctuating terrain -- > (?)

Response :

The manuscript has been revised according to the reviewer's comments.

Figure 6: can you please mark the surface level?

The plot combines topography and spacecraft height. Would it make sense to show topography and spacecraft height separately?

Response :

Figure 6 shows the elevation variations of the lunar terrain along the CE-3 and CE-4 nadir flight traces, which measured from the CE2TMap2015 DEM data. It is not the altitude variation of the detectors. The description in the manuscript has been revised in the manuscript, which makes the meaning of the expression clearer.

L 174

our results reveal that the spacecraft switched from oblique downward to vertical downward...

-- > not clear (it seems that the paper is researching what the spacecraft was doing)

Response :

This study is focused on the reconstruction of the powered descent trajectory of the CE-4, not what the spacecraft was doing. The manuscript has been revised according to the reviewer's comments, which makes the meaning of the expression clearer.

L 176

... which effectively avoid the discontinuity -- > discontinuity? not clear

... which effectively avoid -- > ... which effectively avoided

Response :

The manuscript has been revised according to the reviewer's comments.

L 178

distance measuring sensor -- > can you explain?

General question: were the data from distance measuring sensor and the above mentioned laser imaging sensor used in the reconstruction of the trajectory?

(Note that these instruments are not listed in the "Instrument Description")

Response :

During the process of CE-4 powered descent, sensors including laser ranging sensor, microwave ranging and velocimeter sensor, and an optical imaging sensor and laser imaging sensor were used for on-orbit real-time control. The telemetry data of these sensors was not completely transmitted to the ground and would be not released. Thus, the telemetry data was not used in this study.

Our work is focused on the descent trajectory reconstruction using high-frequency landing sequence images transmitted after safe landing. To avoid making people feel that the paper is researching what the spacecraft was doing, we have revised the manuscript. The detailed process of the CE-4 on-orbit autonomous navigation and the sensors are no longer described, and only the characteristics of the powered descent trajectory reconstruction results are analyzed.

The Automatic on-orbit control technology and sensors including the laser imaging sensor of CE-4 is similar to Chang'E-3 (CE-3). We have added a brief introduction in the manuscript "Discussion" section with the following reference (Li et al., 2016).

Li, S., Jiang, X. Q., Tao, T. Guidance Summary and Assessment of the Chang'e-3 Powered Descent and Landing. JOURNAL OF SPACECRAFT AND ROCKETS. 53 (2): 258-277(2016)

L 186 Based on the sequence images of the LCAM, we reconstructed the power descent trajectory of the CE-4 lander through the photogrammetry method

-- > this is coming very late (in the Discussion?). If I understand correctly, results are already shown earlier in the paper.

Response :

The photogrammetry method which used in the trajectory reconstruction in the study has been mentioned in the introduction section. The detailed information about this method is described in the "Methods" section.

L 191 and L 232: CE2TMap2015 absolute accuracy...

-- > how did the map achieve this excellent accuracy? Unfortunately, the relevant papers that are referenced are all in Chinese.

Response :

The absolute accuracy of CE2TMap2015 has been achieved by global adjustment using CE-2 Stereo images. We added two English references as follows in the manuscript to make the readers understand the detailed process of the method clearer.

*Yan, W., Liu, J., Ren, X., Wang, F., Wang, W., & Li, C. Orbit optimization of Chang'E-2 by global adjustment using images of the moon. *Advances in Space Research*.56(11):2389-2401(2015).*

*Ren, X., Liu, J.J., Li, C.L., et al. A Global Adjustment Method for Photogrammetric Processing of Chang'E-2 Stereo Images. *IEEE Transactions on Geoscience and Remote Sensing*, in press, (2019).*

L 195: Why is the difference in position so large? Does it mean that the LRO reference frame and CE2TMap2015 differ by ~500m? It seems there is some confusion about the reference frame in use.

Response :

The same lunar coordinate system is used for CE2TMap2015 and LRO data, i.e. the mean Earth/polar axis coordinate system. The reference surface of the elevation is the surface of the Moon's spheroid with a radius of 1737.4 km, and the reference origin is the mass center of the Moon.

In line 195 in the manuscript, the planimetric deviation is 433 m between the positions calculated from CE2TMap2015 and LRO data. The deviation is mainly due to the absolute position between these two data. Here, a new reference has been referenced to main text (Ren et al., 2019, in press).

Three-dimensional orbital positions are important input parameter for the lunar terrain reconstruction. There are some system deviation between these orbital positions of CE-2 and LRO because of many reasons, such as the orbit calculation models of CE-2 and LRO, the gravity field model (CE-2 using the LP165P, the LRO using the GRAIL model), the accuracy of the ground radio measurements, and the orbital extrapolation method, etc.

Corresponding, there are some positional deviation between the CE2TMap2015 and the LRO terrain data, which calculated using these orbital positions.

In order to calculate the planimetric deviations between LRO data (GLD100m) and CE2TMap2015, 839 check points uniformly distributed over the entire lunar surface were selected. The results show that the average value was 354m with a standard deviation of 228 m (1 σ). The horizontal position deviation of the CE-4 landing site determined by the two sets of data products is 433 m, which is reasonable within the positional deviation range between the CE2TMap2015 and LRO terrain data.

Ren, X., Liu, J.J., Li, C.L., et al. A Global Adjustment Method for Photogrammetric Processing of Chang'e-2 Stereo Images. *IEEE Transactions on Geoscience and Remote Sensing*, in press, (2019).

Li, C. L., Liu, J. J., Ren, X., et al. Lunar Global High precision Terrain Reconstruction Based on Chang'e-2 Stereo Images. *Geomatics and information Science of Wuhan University*. 43 (4): 486-495 (2018).

L 206

Instrument descriptions -- > This information is coming very late in the paper. The complete list of instrument and relevant data used in the analysis should be briefly introduced earlier in the paper. The details may be moved to the end. It appears that critical instruments are not listed (laser imager, distance sensor?)

Response :

According the format requirement of Nature communication journal, the instrument descriptions should be in the "Methods" section as supporting materials.

During the process of CE-4 powered descent, sensors including laser ranging sensor, microwave ranging and velocimeter sensor, and an optical imaging sensor and laser imaging sensor were used for on-orbit real-time control. The telemetry data of these sensors was not completely transmitted to the ground and would be not released. Thus, the telemetry data was not used in this study.

Our work is focused on the descent trajectory reconstruction using high-frequency landing sequence images transmitted after safe landing. To avoid making people feel that the paper is researching what the spacecraft was doing, we have revised the manuscript. The detailed process of the CE-4 on-orbit autonomous navigation and the sensors are no

longer described, and only the characteristics of the powered descent trajectory reconstruction results are analyzed.

The Automatic on-orbit control technology and sensors including the laser imaging sensor of CE-4 is similar to Chang'E-3 (CE-3). We have added a brief introduction in the manuscript "Discussion" section with the following reference (Li et al., 2016).

Li, S., Jiang, X. Q., Tao, T. Guidance Summary and Assessment of the Chang'e-3 Powered Descent and Landing. JOURNAL OF SPACECRAFT AND ROCKETS. 53 (2): 258-277(2016)

L 249: what is the NAC image used for? It is also not mentioned in Fig.7? Please clarify

Response :

The LRO NAC image is mainly used to assist the selection of GCPs in the adjustment of LCAM. The spatial resolution of the selected 180 LCAM images in this study is between ~5m to 2cm. The spatial resolution of the CE2TMap2015 map is 7m. So, if the spatial resolution of the LCAM images is better than 1m, it is difficult to directly select GCPs from the CE2TMap2015 map. A LRO NAC image (M178833263LC) was selected in this manuscript. It has been precisely rectified using CE2TMap2015 as the base map. The GCPs for the LCAM images, whose resolution is better than 1m, was firstly identified on the rectified LRO NAC image. Then, the planimetric position and elevation was measured from the CE2TMap2015 map. In Fig. 7, we added this processing flow.

L 369 We thank the team ...  perhaps this statement should go into the acknowledgements?

Response :

We accepted the reviewer's suggestion. The sentence "We thank the team ..." has gone into the acknowledgements.

L 374 data ... available on reasonable request  "reasonable" ?

Response :

CE2TMap2015 and CE-4 images are available at Data Publishing and Information Service System of China's Lunar Exploration Program (<http://moon.bao.ac.cn/>). We have revised the description in the "Data availability" section of the manuscript.

Reviewer #3 (Remarks to the Author):

The manuscript submitted by Liu et al presents the photogrammetric methodology utilized to determine

the descent trajectory and surface coordinates of the Chang'e 4 lander.

1) What are the major claims of the paper?

- The authors derived a detailed reconstruction of the descent trajectory, which included autonomous hazard avoidance. They describe how the onboard systems avoided hazards and allowed the vehicle to land safely on the lunar farside. The trajectory and surface coordinate reconstructions were the result of standard photogrammetric bundle block adjustment techniques.

The authors provide the following coordinates for the lander 45.4446°S latitude, 177.5991°E, elevation -5935 meters and state that (lines 199-201) "As a permanent artificial landmark on the farside of the Moon, the location of the CE-4 lander was precisely confirmed using CE-2 and CE-4 image, and can serve as a potential control point on the farside of the Moon."

Absolute coordinate control for the Chang'e 4 images was provided by tying the bundle block adjustment to the Chang'e 2 global image mosaic / control network, with a stated uncertainty of 21-97 m horizontal and 2-19 m elevation. The Chang'e 2 coordinate system is in turn tied to the 5 laser retroreflectors located on the nearside. Note that in the caption to Table 1 a footnote states (line 149): "...the elevation data was accurate to meter." It is not clear to the reviewer where the accurate to a meter was derived when previously it was stated that the uncertainty in elevation was 2-19 m.

Response :

In lines 146-149 of the manuscript, we described the absolute position accuracy of CE2TMap2015, i.e. "Compared to the absolute positions of 5 laser reflectors located on the moon, the horizontal positional deviation of CE2TMap2015 is 21~97 m". With this accuracy we can determine the effective value of the planimetric position and elevation of the CE-4 landing site, instead of the height accuracy to the meter lever.

Considering there is no need to introduce the absolute accuracy of CE2TMap2015 here, and in order to avoid misunderstandings, we removed these contents in the main text, and introduced these contents in the section of "Instruments and dataset descriptions" in methods.

The authors compared their derived coordinates to those derived by the Lunar Reconnaissance Orbiter Camera (LROC) team values (45.457°S, 177.589°E, plus minus 20 meters) and state that there is a 376 m and 215 m difference in latitude and longitude (respectively), for a total 433 m positional offset. The

authors state that: "The total positional deviation is 433 m. Apparently, the positioning data of this study is more sufficient and detailed, and the results are more accurate. The latitude and longitude disagreement with that of LRO data can be accounted for by different reference map." Lines 195-198. [Note: a degree on the Moon is ~30.3 kilometer so 0.001° is ~30 meters]

Response :

As the reviewer mentioned, the distance of 0.001° is equivalent to ~30m on the lunar surface along the longitude direction. But the distance along the latitude direction varies with the latitude value. The latitude of the CE-4 landing site is 45.4446°, where the distance of 0.001° along the latitude direction is equivalent to ~21m. The longitude difference and latitude difference between the LROC team values and our values of CE-4 landing site is 0.0101° and 0.0124° respectively. Corresponding, along the longitude and latitude directions, the planimetric deviations of the CE-4 landing site between the results of this manuscript and the results of LRO team are 376 m and 215 m respectively. So the total planimetric deviation is 433 m.

2) Are they novel and will they be of interest to others in the community and the wider field?

- While the methods presented in this work to derive the trajectory and lander coordinates are not novel, the reconstruction of a robotic lander under autonomy as it sets down on the lunar farside is novel and will be of interest to others in the community and the wider field.

√

3) If the conclusions are not original, it would be helpful if you could provide relevant references.

The conclusions seem to be original, though partially unsubstantiated. Authors might add a standard reference to the bundle block adjustment technique that goes back 60 years (see below).

Response :

We added the following reference in the manuscript according to the reviewer.

Brown, D. C., 1958. A Solution to the General Problem of Multiple Station Analytical Stereo- angulation. Air Force Missile Test Center Report No. 58-8, Patrick AFB, Florida.

4) Is the work convincing, and if not, what further evidence would be required to strengthen the conclusions?

- For the most part the work is convincing. I do have one quibble with the comparison to the LROC team coordinates of the lander summarized above (see also lines 194-198 in manuscript). One could justifiably

argue that the CE-2 coordinate system tied to the 5 nearside retroreflectors will not be as accurate on the opposite side of the Moon for several reasons (i.e. the lumpy gravity field of the Moon, small uncertainties in camera model (distortion, FL) adding up over “distance”, etc.). For example, a control network derived from Clementine UVVIS images was also shown to have km offsets due to the aforementioned reasons (Speyerer et al., 2016).

The authors did not fully document the coordinate system used in their work (a key reference is in Chinese and this reviewer was unable to read the paper) thus the claim that discrepancy is due to “different reference map” is left ambiguous. The LROC coordinates derived have a stated accuracy much smaller than the 433 meters reported in this manuscript.

Lunar Reconnaissance Orbiter radiometric tracking data were combined with the GRAIL gravity model to significantly improve the Lunar Reconnaissance Orbiter spacecraft ephemeris and thus geodetic accuracy of each LOLA spot to less than 10 m horizontally and 1 m vertically (Lemoine et al. 2014; Mazarico et al. 2012; Mazarico et al. 2013). Subsequent, in-flight calibration has improved the mapping of a single NAC image to < 20 m (Speyerer et al., 2016). The methods used to control LROC NAC images to the LOLA framework (and associated uncertainties) are documented in Wagner et al (2017) and Henriksen et al (2017).

Archinal, B.A., M.R. Rosiek, R.L. Kirk, and B.L. Redding, 2006. The Unified Lunar Control Network 2005. U.S. Geological Survey Open-File Report 2006-1367, 12 p.

Henriksen, M. R, M. R. Manheim, K. N. Burns, P. Seymour, E. J. Speyerer, A. Deran, A. K. Boyd, E. Howington-Kraus, M. R. Rosiek, B. A. Archinal, and M. S. Robinson (2017), Extracting accurate and precise topography from LROC narrow angle camera stereo observations, *Icarus*, 283, 122-137, doi: 10.1016/j.icarus.2016.05.012.

Lemoine, F.G., Goossens, S., Sabaka, T.J., et al., 2014. GRGM900C: A degree 900 lunar gravity model from GRAIL primary and extended mission data. *Geophys. Res. Lett.* 41, 3382–3389. doi: 10.1002/2014GL060027.

Mazarico, E., Rowlands, D.D., Neumann, G.A., et al., 2012. Orbit determination of the lunar reconnaissance orbiter. *J. Geod.* 86 (3), 193–207. doi: 10.1007/s00190-011-0509-4.

Mazarico, E., Goossens, S.J., Lemoine, F.G, et al. , 2013. Improved orbit determination of lunar orbiters with lunar and gravity fields obtained by the GRAIL mission. In: Proceedings of the 44th Lunar and Planetary Science Conference. The Woodlands, TX, Abstract #2414.

Speyerer, E.J., Wagner, R.V., and Robinson, M.S., 2016. Geometric Calibration of the Clementine UVVIS Camera Using Images Acquired by the Lunar Reconnaissance Orbiter. In: The International Archives of the Photogrammetry, Remote Sensing and Spatial Information Sciences, Volume XLI-B4

Speyerer, E.J., R.V. Wagner, M.S. Robinson, A. Licht, P.C. Thomas, K. Becker, J. Anderson, S.M. Brylow, D.C. Humm, M. Tschimmel, 2016. Pre-flight and On-orbit Geometric Calibration of the Lunar Reconnaissance Orbiter Camera, Space Science Reviews, 200, 357-392.

Wagner, R. V., D. M. Nelson, J. B. Plescia, M. S. Robinson, E. J. Speyerer, and E. Mazarico (2017), Coordinates of anthropogenic features on the Moon, Icarus, 283, 92-103, doi: 10.1016/j.icarus.2016.05.011.

Yes, different reference maps were used: LOLA vs Chang'e 2 for the LROC vs Chang'e 4 work. The manuscript does not in any way document the assertion: "Apparently, the positioning data of this study is more sufficient and detailed, and the results are more accurate." The authors need to compare the Chang'e 2 coordinate system with the LOLA system in a global sense and make some quantitative assessment of any discrepancies between the two products to support the statement quoted above (lines 195-196 in the manuscript).

Response :

The results of (Yan et al, 2015; Li et al., 2018; Ren et al, 2019) show that the average planimetric and height deviations between neighboring strips of CE2TMap2015 were 5 m and 2 m (<1pixel, with a spatial resolution of 7 m). (Lemoine et al. 2014; Mazarico et al. 2012; Mazarico et al. 2013) solved the LRO orbit data combining with LRO ground tracking data and GRAIL data. The planimetric and height accuracy are less than 10m and 1m, respectively. The accuracy of LRO LOLA lunar footprint coordinates calculated by the orbit data also reaches this level.

In order to calculate the planimetric deviations between LRO GLD100m and CE2TMap2015, 839 check points uniformly distributed over the entire lunar surface were selected. The results show that the average planimetric deviation was 354 m with a standard deviation of 228 m (1 σ) (Li et al., 2018; Ren et al, 2019).

It can be seen that CE2TMap2015 and LRO data both have a good inner consistency. However, as the reviewer mentioned, there is the planimetric deviation of 354m with a standard deviation of 228 m (1 σ) between these two data because of many reasons, such as the orbit calculation models of CE-2 and LRO, the gravity field model (CE-2 using the LP165P, the LRO using the GRAIL model), the accuracy of the ground radio

measurements, and the orbital extrapolation method, etc. Corresponding, the horizontal position deviation of the CE-4 landing site determined by the two sets of data products is 433 m, which is reasonable within the positional deviation range between the CE2TMap2015 and LRO terrain data.

Therefore, the position of the CE-4 landing site determined in this study is authentic, which can reveal the actual position on the lunar farside. At the same time, in order to easily find the position of the CE-4 landing site, the pixel coordinates of the landing site were given in multiple LCAM images, which will be convenient for subsequent applications.

In lines 195-196 of the manuscript, we change old description (“obviously, the positioning data of this study is more complete and detailed, the results are more accurate.”) to “The position of the CE-4 landing site determined in this study is authentic, which can reveal the actual location on the lunar farside. In this study, the CE-4 descent trajectory was recovered, and the landing site position was accurately determined. The pixel coordinates of the landing site were given in multiple LCAM images”.

Reference:

Yan, W., Liu, J., Ren, X., Wang, F., Wang, W., & Li, C. Orbit optimization of Chang'E-2 by global adjustment using images of the moon. *Advances in Space Research*.56(11):2389-2401(2015).

Ren, X., Liu, J.J., Li, C.L., Li, H.H., Wang, F.F., Yan, W., Wang, W.R., Zhang, X.X., Gao, X.Y & Chen, W.L. A Global Adjustment Method for Photogrammetric Processing of Chang'E-2 Stereo Images. *IEEE Transactions on Geoscience and Remote Sensing*, in press 2019.

Li, C. L., Liu, J. J., Ren, X., et al. Lunar Global High precision Terrain Reconstruction Based on Chang'e-2 Stereo Images. *Geomatics and information Science of Wuhan University*. 43 (4): 486-495 (2018).

Lemoine, F.G., Goossens, S., Sabaka, T.J., et al.. GRGM900C: A degree 900 lunar gravity model from GRAIL primary and extended mission data. *Geophys. Res. Lett.* 41, 3382–3389. doi: 10.1029/2014GL060027 (2014).

Mazarico, E., Rowlands, D.D., Neumann, G.A., et al., Orbit determination of the lunar reconnaissance orbiter. *J. Geod.* 86 (3), 193–207. doi: 10.1007/s00190-011-0509-4 (2012).

Mazarico, E., Goossens, S.J., Lemoine, F.G, et al. Improved orbit determination of lunar orbiters with lunar and gravity fields obtained by the GRAIL mission. In: *Proceedings of the 44th Lunar and Planetary Science Conference*. The Woodlands, TX, Abstract #2414 (2013).

5) On a more subjective note, do you feel that the paper will influence thinking in the field?

The description of the descent trajectory and derivation of the coordinates is certainly of broad interest, however I do not think the manuscript will influence thinking in the field.

✓

General Comments To Authors

1) There are a few instances of inappropriate word usage and faulty grammar, likely due to translation to English. Some logical inconsistencies may wholly or partially be due to a language issue.

Examples:

1) Line 15 16: "Because it is impossible to directly measure the farside of the Moon, we cannot acquire the descent trajectory and the landing site of the spacecraft." It is possible to directly measure the lunar farside and the Chinese L2 relay satellite Queqiao should have been able to receive telemetry during the descent I cannot understand what is meant by this sentence. I can guess, but the authors should clarify. Perhaps the lander did not have the ability to transmit to Queqiao during the descent?

Response :

There was no radio measurement equipment between the CE-4 and the relay satellite Queqiao, it was unable to perform radio measurements by the ground tracking network on the lunar farside.

As the reviewer mentioned, during the CE-4 powered descent, real-time telemetry data (including altitude, acceleration, attitude, etc) can be obtained by the relay satellite Queqiao. The telemetry data is mainly used for on-orbit real-time control. But the downlinked telemetry data was very limited because the procedure of the powered descent was short and speed changed rapidly. And these data will be not released. So, the telemetry data was not used in this article. The localization technology based on the LCAM images was used to reconstruct the CE-4 lander descent trajectory and to confirm the precise landing site location after CE-4 landing.

In order to avoid misunderstandings, we have revised the relevant content in the main text, as following:

"... It is almost the only effective way to reconstruct the landing trajectory and determine the location of the landing site using a series of image data obtained during descent when there were no Earth-based radio measurements and the telemetry data."

2) Please spell out terms before using abbreviations/acronyms (i.e. DOM, DEM)

Response :

The full name of DOM and DEM is added in the manuscript, i.e. Digital Orthophoto Map and Digital Elevation Model.

3) "farside of the Moon" appears multiple times. After the first instance you can drop the "of the Moon" and just say farside.

4) "5m/p, 1m/p, 10cm/p, 5cm/p resolution" appears multiple times throughout the manuscript. Once or twice is plenty.

Response :

The manuscript has been revised according to the reviewer's comments.

5) The values in Table 4 and 5 (and elsewhere in the manuscript) may be overly precise. Are three decimal places really justifiable (mm precision)?

Response :

The highest resolution of the LCAM images selected in this study is ~2mm where near the landing site. The resolution of the NCAM images selected here is between 1mm to 8mm. So the effective number of control point errors after adjustment is retained to the millimeter. Correspondingly, in Tables 4 and 5, the effective number of the planimetric and height errors is also retained to the millimeter or three decimals.

Consider adding Classic Bundle Block Adjustment References:

Schmid, H.H., 1958, Eine allgemeine analytische Lösung für die Aufgabe der Photogrammetrie,

Bildmessung und Luftbildwesen 1958, pp.103-113, 1959 pp.1-12. [German]

Brown, D. C., 1958. A Solution to the General Problem of Multiple Station Analytical Stereo- angulation.

Air Force Missile Test Center Report No. 58-8, Patrick AFB, Florida. [English]

Response :

We added the following reference in the manuscript according to the reviewer.

Brown, D. C., 1958. A Solution to the General Problem of Multiple Station Analytical

Stereo- angulation. Air Force Missile Test Center Report No. 58-8, Patrick AFB, Florida.

Reviewers' comments:

Reviewer #1 (Remarks to the Author):

I had no issues with the updated paper.

Reviewer #2 (Remarks to the Author):

Review of Paper

„Descent Trajectory reconstruction and landing site positioning of Chang'E-4 on Moon's farside“ by Jianjun Liu and Co-authors

Again, this is a most interesting report on the first historic landing on the lunar far-side, quite appropriate for publication in Nature.

The authors have responded very effectively on the comments by the referees. Also, wording and phrasing has very much improved.

I only have minor comments and remaining requests for clarifications. I recommend publication of the exciting material, pending on these clarifications.

Specifically:

Abstract

series of image data -- > series of images

radio measurements -- > radio tracking

using photogrammetric processed images -- > using photogrammetrically processed images -- > (or) using photogrammetric image processing techniques

It is almost the only effective way to reconstruct ... -- > We effectively reconstruct ...

Fig. 1 d and e: please label the axis (m?). Please mark the North direction (or say in the caption). Note: There is no need to give the scale bar (0, 100, 200 m) when the relevant data are obvious from the axes.

Fig. 4b: numbers on the color-bar: Shouldn't the elevation be km?

Considering the elevation profile given in Fig 6: wouldn't it be useful to show the approach trajectory on Fig. 4b?

Fig. 4c and d: Judging from the size of the rectangles, shouldn't the resolution of 4c be better than the resolution of 4d? The caption says it the other way.

Besides, craters in Fig 4c are difficult to relate to Figures 4d,e, or f.

Please check numbers on scale bars. Shouldn't the scale bar in 4c be in km?

Fig. 5 b. It is not clear, where the North arrow is pointed in this perspective viewing. Looking at Fig. 5a: From where was the image 5b taken? Which one is the crater in the background seen in 5b?

Fig. 5 caption

There are 5 craters around the landing site -- > Five prominent craters around the landing site are marked

Perhaps, one should add before the last sentence: Prominent craters are marked, as in (a).

(Note that Fig 5a is almost identical to Fig. 3d)

The landing position was determined according to (a), and the position of the lander's leg was judged by (b).

-- >

While landing position was determined from orbital data (a), the orientation of the lander was determined using images taken from the ground (b).

Text above Fig. 6:

the lander attitude was adjusted to a vertical attitude (?)

-- > the lander attitude was adjusted to enable thrusting for near-vertical descent

Please add: Elevation is given with respect to ... (?)

Figure 7 caption

The designed trajectories -- > "designed" ?

The dotted border -- > The marked rectangle

The reconstruction trajectories -- > Magnified portion of the reconstructed trajectory

Please add: Altitude is given with respect to final landing site level.

Page 12 and Fig. 9:

Perhaps, it is better to say: stones -- > rocks

Fig. 9: what sizes are the rocks?

Reviewer #3 (Remarks to the Author):

The authors have not adequately responded to the biggest issue: Their implication that the 433 meter discrepancy between the manuscript coordinates and the LROC team coordinates for the Chang'e 4 lander can be explained by a more accurate Chang'e 4 derivation (and other associated mission data).

Text from the rebuttal.

"Therefore, the position of the CE-4 landing site determined in this study is authentic, which can reveal the actual position on the lunar farside. At the same time, in order to easily find the position of the CE-4 landing site, the pixel coordinates of the landing site were given in multiple LCAM images, which will be convenient for subsequent applications.

In lines 195-196 of the manuscript, we change old description ("obviously, the positioning data of this study is more complete and detailed, the results are more accurate.") to "The position of the CE-4 landing site determined in this study is authentic, which can reveal the actual location on the lunar farside. In this study, the CE-4 descent trajectory was recovered, and the landing site position was accurately determined. The pixel coordinates of the landing site were given in multiple LCAM images."

From the revised manuscript line 203: "...and the landing site position was accurately determined."

[Note – I do not understand the use of the word authentic in the rebuttal and the revised manuscript, I think this might be a translation issue?]

- The authors state that the absolute uncertainty of their reference frame is "decameter level" (line 312). (Reference 21 says "21-97 m" as does the methods section.)

- The reviewer originally pointed out that the LROC frame of reference uncertainty is about 20 meters (though the manuscript under review does not mention that fact). See the papers mentioned in the original review that document the coordinate framework for LRO (Lemoine et al. 2014; Mazarico et al. 2012; Mazarico et al. 2013) and LROC (Speyerer et al., 2016, Wagner et al 2017, Henriksen et al 2017).

- The authors acknowledge that the LROC team coordinates disagree by 433 m from their estimate (line 200), and state that this is "within the positional deviation range between the CE2TMap2015 and LRO terrain data [20,21] (see Methods section for details)." (Note: The Methods section says nothing about the LRO coordinate reference frame.)

http://ch.whu.edu.cn/EN/volumn/volumn_1422.shtml

- Reference 21 (http://ch.whu.edu.cn/EN/volumn/volumn_1422.shtml) does seem to be the source of their statement that the LRO and Chang'2 laser coordinates agree within uncertainty. Reference 21 also has a figure (pasted below) that shows that either the CE2TMap2015 or the GLD100/LOLA DEMs are grossly in error on the farside, disagreeing by ~500 m horizontally over the CE-4 landing site (and at least that much over most of the much of

the farside). Neither reference 21 nor the paper under review give any rationale for why they assume that CE2TMap2015 is more accurate. Note that Reference 20 is in press and not available to this reviewer.

图 3 CE-2 DEM 和 LOLA DEM 高程偏差分布图

Fig.3 Distribution Map of Elevation Deviation Between CE-2 DEM and LOLA DEM

图 4 CE-2 DOM 与 GLD100 DOM 平面位置偏差分布图

Fig.4 Distribution Map of Horizontal Position Deviation Between CE-2 DOM and GLD100 DOM

Figure 3 and 4 from Li, C. L., Liu, J. J., Ren, X., et al. Lunar Global High precision Terrain Reconstruction Based on Chang'e-2 Stereo Images. *Geomatics and information Science of Wuhan University* . 43 (4): 486-495 (2018). Note that Google Translate was used to help this non-Chinese reading reviewer read the publication.

Bottom line: The authors have not adequately responded to the biggest issue: Their implication that the 433 meter discrepancy between the manuscript coordinates and the LROC team coordinates for the Chang'e 4 lander can be explained by a more accurate Chang'e 4 derivation (and other associated mission data).

We sincerely thank the reviewers for their critical reading of our manuscript, which help to significantly improve our manuscript. We addressed all remarks made by the 3 reviewers. Our responses are given in red below each remark.

Reviewers' comments:

Reviewer #1 (Remarks to the Author):

I had no issues with the updated paper.

✓

Reviewer #2 (Remarks to the Author):

Review of Paper

Descent Trajectory reconstruction and landing site positioning of Chang'E-4 on Moon's farside" by Jianjun Liu and Co-authors

Again, this is a most interesting report on the first historic landing on the lunar far-side, quite appropriate for publication in Nature.

The authors have responded very effectively on the comments by the referees. Also, wording and phrasing has very much improved.

I only have minor comments and remaining requests for clarifications. I recommend publication of the exciting material, pending on these clarifications.

Specifically:

Abstract

series of image data -- > series of images

radio measurements -- > radio tracking

using photogrammetric processed images -- > using photogrammetrically processed images -- > (or) using photogrammetric image processing techniques

It is almost the only effective way to reconstruct ... -- > We effectively reconstruct ...

Response :

The manuscript has been revised according to the reviewer's comments.

Fig. 1 d and e: please label the axis (m?). Please mark the North direction (or say in the caption).
Note: There is no need to give the scale bar (0, 100, 200 m) when the relevant data are obvious from the axes.

Response :

The axis is labeled with unit "m" and the North direction is marked in Fig. 1d and e. The scale bar is deleted in these figures.

Fig. 4b: numbers on the color-bar: Shouldn't the elevation be km?

Considering the elevation profile given in Fig 6: wouldn't it be useful to show the approach trajectory on Fig. 4b?

Response :

The elevation is in meters in Fig. 4b. The approach trajectory is added in Fig. 4a and b according to the reviewer's comments.

Fig. 4c and d: Judging from the size of the rectangles, shouldn't the resolution of 4c be better than the resolution of 4d? The caption says it the other way.

Besides, craters in Fig 4c are difficult to relate to Figures 4d,e, or f.

Response :

The white rectangle in Fig. 4c represents the border of Fig. 4d. Because Fig. 4c is obtained at a higher altitude than Fig. 4d, the resolution of 4d is better than the resolution of 4c. We have revised the textual description of the manuscript to make it more clearly. "and the white borders in (c), (d), and (e) represent the positions of (d), (e), and (f) in the images, respectively." is revised to "and the white rectangles in (c), (d), and (e) represent the borders of (d), (e), and (f), respectively."

Because Fig. 4c is obtained at a higher altitude than Fig. 4d, 4e, 4f, it can cover larger areas. Coverage area of Fig. 4c, 4d, 4e, 4f are 5km×5km, 1km×1km, 100m×100m, 50m×50m, respectively. The features of the lunar surface expressed in these figures will be significantly different.

Please check numbers on scale bars. Shouldn't the scale bar in 4c be in km?

Response :

Numbers on scale bars in Fig. 4c is revised. The scale bar is in meters.

Fig. 5 b. It is not clear, where the North arrow is pointed in this perspective viewing. Looking at Fig. 5a: From where was the image 5b taken? Which one is the crater in the background seen in 5b?

Response :

The lander side with red flag faces to North in this perspective viewing. Figure 5b is obtained by panoramic camera (PCAM) when the Yutu-2 rover moved to the northwest of the lander. In the revised manuscript, the location of Yutu-2 rover is marked as “#” in Fig. 5a and as a 3D model in Fig. 5d (original Fig.5c). The distance between the rover and the lander is about 18m at that time. Since the 5 craters marked in Figures 5a and 5d are far from the lander (more than 20m), the craters can't be seen in the PCAM's field of view.

CE-4 landing site can be also located in the Lunar Reconnaissance Orbiter (LRO) narrow angle camera (NAC) images, whose relative position to the surrounding terrain is consistent with our results. LRO NAC image of the CE-4 landing site collected on 1 February 2019 (M1303640934LR) is added as Fig.5c.

Fig. 5 caption

There are 5 craters around the landing site -- > Five prominent craters around the landing site are marked

Perhaps, one should add before the last sentence: Prominent craters are marked, as in (a).

(Note that Fig 5a is almost identical to Fig. 3d)

Response :

The manuscript has been revised according to the reviewer's comments.

Fig. 5a and Fig. 3d are for different uses. Fig 5a is used to describe the lunar surface features around the lander, in which 5 typical craters around CE-4 landing site are marked. The location of Yutu-2 rover is also marked in the revised manuscript. Fig. 3d is used to describe the location of CE-4 landing site on the DOM produced by landing camera images.

The landing position was determined according to (a), and the position of the lander's leg was judged by (b).-- > While landing position was determined from orbital data (a), the orientation of the lander was determined using images taken from the ground (b).

Response :

The manuscript has been revised according to the reviewer's comments.

Text above Fig. 6:

the lander attitude was adjusted to a vertical attitude (?)

-- > the lander attitude was adjusted to enable thrusting for near-vertical descent

Response :

The manuscript has been revised according to the reviewer's comments.

Please add: Elevation is given with respect to ... (?)

Response :

The manuscript has been revised according to the reviewer's comments. "Elevation is given with respect to the Moon's spheroid with a radius of 1737.4 km." is added in the revised manuscript.

Figure 7 caption

The designed trajectories -- > "designed" ?

The dotted border -- > The marked rectangle

The reconstruction trajectories -- > Magnified portion of the reconstructed trajectory

Response :

The manuscript has been revised according to the reviewer's comments. The trajectories in Fig. 7 (a) are indeed designed ones, which used to show the entire powered descent. The trajectory reconstructed in this paper is only the part below the altitude of 6km as showed in Fig. 7 (b).

Please add: Altitude is given with respect to final landing site level.

Response :

The manuscript has been revised according to the reviewer's comments. "Altitude is given with respect to final landing site level." is added in the revised manuscript.

Page 12 and Fig. 9:

Perhaps, it is better to say: stones → rocks

Fig. 9: what sizes are the rocks?

Response :

The manuscript has been revised according to the reviewer's comments. Sizes of the rocks range from a few centimeters to a dozen centimeters. The scale bar is added in the revised manuscript.

Reviewer #3 (Remarks to the Author):

The authors have not adequately responded to the biggest issue: Their implication that the 433 meter discrepancy between the manuscript coordinates and the LROC team coordinates for the Chang'e 4 lander can be explained by a more accurate Chang'e 4 derivation (and other associated mission data).

Text from the rebuttal.

“Therefore, the position of the CE-4 landing site determined in this study is authentic, which can reveal the actual position on the lunar farside. At the same time, in order to easily find the position of the CE-4 landing site, the pixel coordinates of the landing site were given in multiple LCAM images, which will be convenient for subsequent applications.

In lines 195-196 of the manuscript, we change old description (“obviously, the positioning data of this study is more complete and detailed, the results are more accurate.”) to “The position of the CE-4 landing site determined in this study is authentic, which can reveal the actual location on the lunar farside. In this study, the CE-4 descent trajectory was recovered, and the landing site position was accurately determined. The pixel coordinates of the landing site were given in multiple LCAM images.”

From the revised manuscript line 203: “...and the landing site position was accurately determined.”

[Note – I do not understand the use of the word authentic in the rebuttal and the revised manuscript, I think this might be a translation issue?]

- The authors state that the absolute uncertainty of their reference frame is "decameter level" (line 312). (Reference 21 says "21-97 m" as does the methods section.)

- The reviewer originally pointed out that the LROC frame of reference uncertainty is about 20 meters (though the manuscript under review does not mention that fact). See the papers mentioned in the original review that document the coordinate framework for LRO (Lemoine et al. 2014; Mazarico et al. 2012; Mazarico et al. 2013) and LROC (Speyerer et al., 2016, Wagner et al 2017, Henriksen et al 2017).

- The authors acknowledge that the LROC team coordinates disagree by 433 m from their estimate (line 200), and state that this is "within the positional deviation range between the CE2TMap2015 and LRO terrain data [20,21] (see Methods section for details)." (Note: The Methods section says nothing about the LRO coordinate reference frame.)

http://ch.whu.edu.cn/EN/volumn/volumn_1422.shtml

- Reference 21 (http://ch.whu.edu.cn/EN/volumn/volumn_1422.shtml) does seem to be the source of their statement that the LRO and Chang'2 laser coordinates agree within uncertainty. Reference 21 also has a figure (pasted below) that shows that either the CE2TMap2015 or the GLD100/LOLA DEMs are grossly in error on the farside, disagreeing by ~500 m horizontally over the CE-4 landing site (and at least that much over most of the much of the farside). Neither reference 21 nor the paper under review give any rationale for why they assume that CE2TMap2015 is more accurate. Note that Reference 20 is in press and not available to this reviewer.

图3 CE-2 DEM 和 LOLA DEM 高程偏差分布图

Fig.3 Distribution Map of Elevation Deviation Between CE-2 DEM and LOLA DEM

图4 CE-2 DOM 与 GLD100 DOM 平面位置偏差分布图

Fig.4 Distribution Map of Horizontal Position Deviation Between CE-2 DOM and GLD100 DOM

Response :

We sincerely thank the reviewer for his comments. In this study, the location of CE-4 landing site was confirmed on the basemap CE2TMap2015 using photogrammetrically processed images of CE-4 landing camera and navigation camera. CE-4 landing site can be also located in the Lunar Reconnaissance Orbiter (LRO) narrow angle camera (NAC) images, whose relative position to the surrounding terrain is consistent with our results. LRO NAC image of the CE-4 landing site collected on 1 February 2019 (M1303640934LR) is added as Fig.5c.

Our coordinates disagree by more than 400 meters from the LROC team coordinates for the CE-4 lander, which is an objective reflection of the deviation between the two sets of terrain data on the lunar farside. However, it cannot be inferred that the coordinates of CE-4 landing site based on CE2TMap2015 is more accurate (see below for details). Therefore, the description like "...and the landing site position was accurately determined.", "authentic" has been removed from the revised manuscript. Furthermore, the significance of establishing absolute control points on the lunar farside for future lunar exploration missions is discussed. In the method section, the description of LRO terrain data uncertainty and related references are mentioned to avoid the illusion that the absolute coordinates of the CE-4 landing site based on CE2TMap2015 is more accurate.

It should be noted that there are differences in the data processing methods and processes between CE2TMap2015 and LRO terrain data. CE2TMap2015 used 5 absolute control points located on the nearside as position constraints to perform global adjustment, optimized measurement data such as orbit ephemeris and spacecraft attitude, and then obtained the lunar surface position. LRO terrain data is based on a frame of reference that jointed radio tracking data and LOLA data combined with the GRAIL gravity model to improve orbital measurement data, and then obtained the lunar surface position. It can be seen that the two sets of terrain data have their own characteristics.

(1) Relative positioning accuracy

The average horizontally and vertically deviations between neighboring strips of CE2TMap2015 were 5 m and 2 m (<1 pixel, with a spatial resolution of 7 m = after global adjustment (Yan et al, 2015; Li et al., 2018; Ren et al, 2019). In contrast, LRO ephemeris accuracy is less than 10 m horizontally and 1 m vertically by combining radiometric tracking data and LOLA data with the GRAIL gravity model (Lemoine et al. 2014; Mazarico et al. 2013). It can be seen that both CE2TMap2015 and LRO data have good internal consistency.

(2) Absolute positioning accuracy

On the lunar nearside, 5 absolute control points can be used to assess the absolute positioning accuracy of the two set of terrain data. Compared to the positions of the 5 absolute control points, the horizontal positional deviation of CE2TMap2015 is 21~97 m [20,21]. The LROC frame of reference uncertainty is about 20 meters (Speyerer et al.,

2016) and the positional deviation from the 5 absolute control points is < 5.2 m (Wagner et al 2017) .

On the lunar farside, however, it is difficult to compare the absolute positioning accuracy of two sets of terrain data because there are no absolute control points. CE-2 coordinate system tied to the 5 nearside retroreflectors will not be as accurate on the opposite side of the Moon. Discussion about CE2TMap2015 absolute positional accuracy on the farside has been removed from the revised manuscript. The positioning deviation between CE2TMap2015 and LRO terrain data on the lunar farside will both increase due to several reasons, such as the orbital measurement error, the lumpy gravity field of the Moon on the lunar farside, small uncertainties in camera model (distortion, FL), etc.. The positioning accuracy for the two sets of terrain data on the farside may be worse than that on the nearside, and Reference [20,21] show that the positional deviation between them will become larger. Compared to the LRO terrain data, CE2TMap2015 shows the global average positional deviation of 354m, with a standard deviation of 228 m (1 σ).

The revised manuscript shows a discrepancy of 415 meters between our coordinates (177.5991°E, 45.4446°S, -5935 m) and the LROC team coordinates (177.5885°E, 45.4561°S, -5927m) for the CE-4 landing site, which is within the positional deviation range mentioned above and reflects the deviation of the two sets of terrain data on the lunar farside (Co-contribution). Our work does not discuss the absolute positioning accuracy of the two sets of terrain data. The revised manuscript also does not imply whose accuracy of the positioning of the two sets of terrain data on the lunar farside is better. In order to effectively eliminate this deviation, in the discussion section of the revised manuscript, we point out it is a feasible method to establish absolute control points on the lunar farside, which can be an effort in the future lunar exploration program.

[NOTE – We used new coordinates from LROC team (<http://www.lroc.asu.edu/posts/1100>, on April 30) in the revised manuscript instead of the original ones (177.589°E, 45.457°S) (<http://www.lroc.asu.edu/posts/1087>, on January 11)]

The above analyses and opinions have been reflected in the discussion and methods sections of the revised manuscript.

(1) In the Discussion section,

“The position of the CE-4 landing site determined in this study is authentic, which can reveal the actual location on the lunar farside. In this study, the CE-4 descent trajectory was recovered, and the landing site position was accurately determined. The pixel coordinates of the landing site were given in multiple LCAM images.”is deleted.

On the other hand, “Compared with the positioning results of the landing site based on LRO terrain data (177.5885°E, 45.4561°S, -5927m)²⁴, our results show a 226 m deviation along the latitude direction and a 348 m deviation along the longitude direction. The total positional deviation is 415m, which reflects the deviation of the two sets of terrain data on the lunar farside (see the Methods section for details). A feasible method to effectively

eliminate this deviation is to establish absolute control points on the lunar farside.” is supplemented.

(2) In the Methods section, the following analysis is supplemented.

” LRO terrain data was used for CE-4 landing site position comparison. Orbit overlap analyses show that, LRO spacecraft ephemeris can be improved to less than 10 m horizontally and 1 m vertically by combining radiometric tracking and LOLA data with the GRAIL gravity model²⁵⁻²⁷. As a result, the LRO terrain data uncertainty is about 20 meters²⁸ and the positional deviation from the 5 laser reflectors is $< 5.2 \text{ m}^{29}$. A pair of LRO NAC observations of the CE-4 landing site were collected on 1 February 2019 (M1303619844LR, M1303640934LR), and were used to create a digital terrain model (DTM) on the LOLA-plus-GRAIL coordinate framework. The coordinates of CE-4 lander is estimated to be (177.5885°E, 45.4561°S, -5927m) ²⁴ based on the DTM.

It can be seen that both CE2TMap2015 and LRO terrain data show good internal consistency. However, the positioning deviation of the two sets of terrain data on the lunar farside will both increase due to several reasons, such as orbital measurement error, the lumpy gravity field of the Moon on the lunar farside, small uncertainties in camera model (distortion, FL), etc.. Compared to the LRO terrain data, the average global positional deviation of CE2TMap2015 is 354m with a standard deviation of 228 m (15)^{20,21,30}, which mainly results from large farside deviation of two sets of terrain data. The 415 m discrepancy between our coordinates and the LRO terrain data coordinates for the Chang’e 4 lander is within the positional deviation range.”

(3) “decimeter level” for absolute positional accuracy of CE-4 landing site based on CE2TMap2015 in Table 1 and Line312 has been deleted in the revised manuscript.

(4) New coordinates (177.5885°E, 45.4561°S, -5927m) from LROC team for the CE-4 landing site are used in the revised manuscript.

(5) Reference 20 has been available for access now (doi:10.1109/TGRS.2019.2908813). Reviewer can acknowledge the production process and accuracy analysis of CE2TMap2015 through it. In addition, we also provide a pdf file of this reference for you.

Ren, X., Liu, J.J., Li, C.L., et al. A Global Adjustment Method for Photogrammetric Processing of Chang'E-2 Stereo Images. IEEE Transactions on Geoscience and Remote Sensing. (2019) doi: 10.1109/TGRS.2019.2908813